# Rapid and site-specific deep phosphoproteome profiling by data-independent acquisition without the need for spectral libraries

Dorte B. Bekker-Jensen [1], Oliver M. Bernhardt[2], Alexander Hogrebe [1], Ana Martinez-Val [1], Lynn Verbeke[2], Tejas Gandhi [2], Christian D. Kelstrup [1], Lukas Reiter [2] & Jesper V. Olsen [1]*

Quantitative phosphoproteomics has transformed investigations of cell signaling, but it remains challenging to scale the technology for high-throughput analyses. Here we report a rapid and reproducible approach to analyze hundreds of phosphoproteomes using data-independent acquisition (DIA) with an accurate site localization score incorporated into Spectronaut. DIA-based phosphoproteomics achieves an order of magnitude broader dynamic range, higher reproducibility of identification, and improved sensitivity and accuracy of quantification compared to state-of-the-art data-dependent acquisition (DDA)-based phosphoproteomics. Notably, direct DIA without the need of spectral libraries performs close to analyses using project-specific libraries, quantifying > 20,000 phosphopeptides in 15 min single-shot LC-MS analysis per condition. Adaptation of a 3D multiple regression model-based algorithm enables global determination of phosphorylation site stoichiometry in DIA. Scalability of the DIA approach is demonstrated by systematically analyzing the effects of thirty kinase inhibitors in context of epidermal growth factor (EGF) signaling showing that specific protein kinases mediate EGF-dependent phospho-regulation.

[1] Novo Nordisk Foundation Center for Protein Research, Proteomics Program, Faculty of Health and Medical Sciences, University of Copenhagen, Blegdamsvej 3b, 2200 Copenhagen, Denmark. [2] Biognosys AG, Wagistrasse 21, 8952 Schlieren, Switzerland. *email: jesper.olsen@cpr.ku.dk

Site-specific protein phosphorylation is one of the most important post-translational modifications (PTMs) as it can rapidly modulate a protein's function by changing its activity, subcellular localization, interactions, or stability[1]. It is a highly dynamic modification that regulates essentially all cellular signaling networks. Deregulated phospho-signaling is therefore a hallmark of cancer and many other diseases. Major advances in phosphopeptide enrichment strategies, instrument performance, and computational analysis tools have made mass spectrometry-based phosphoproteomics the method of choice for the study of protein phosphorylation on a global scale. Large-scale quantitative phosphoproteomics has proven to be successful in addressing unsolved questions in cell signaling and biomedicine[2–5]. However, the majority of successful phosphoproteomics studies typically involves days or even weeks of measurements by liquid chromatography tandem mass spectrometry (LC-MS/MS) to analyze few cellular conditions with sufficient depth to identify and pinpoint the functional phosphorylation sites. Moreover, current tandem mass spectrometric sequencing speed and the semi-stochastic nature of data-dependent acquisition (DDA) make it challenging for phosphoproteomics to systematically and reproducibly analyze phosphorylation sites across large numbers of samples. This limits its application for high-throughput applications such as drug screening.

With the advent of fast scanning high-resolution tandem mass spectrometers, DIA has appeared as a powerful alternative to DDA in shotgun proteomics[6–8]. In a DIA analysis, all (phospho)peptides within a predefined mass-to-charge ($m/z$) window are co-fragmented and the resulting fragments measured together. This analysis is repeated as the mass spectrometer goes through the full mass range, which facilitates systematic measurement of all peptide ions regardless of their intensity and overcomes the precursor selection problem of DDA. DIA typically provides broader dynamic range, higher peptide identification rates, improved reproducibility of identification, and accuracy for quantification. However, due to the multiplexed fragment ion spectra, DIA requires more elaborate data processing algorithms and software solutions for spectral deconvolution, which typically make use of pre-recorded spectral libraries. Moreover, apart from the unambiguous identification of the phosphopeptide sequence, the deconvoluted tandem mass spectra should contain sufficient information to localize phosphorylation sites with single amino acid resolution[1].

To address these issues, we here develop an optimized label-free quantitative phosphoproteomics approach combining fast liquid chromatography tandem mass spectrometry with site-specific data-independent acquisition (DIA). This approach allows us to systematically and reproducibly analyze more than 10,000 phosphorylation sites across hundreds of samples. Furthermore, we develop and employ algorithms to accurately localize phosphorylation sites in DIA datasets and determine their fractional stoichiometry on a system-wide scale. We apply this strategy to identify phosphorylation site targets of ten major protein kinases in the epidermal growth factor signaling pathway[9].

## Results

**Comparison of DDA and DIA for quantitative phosphoproteomics**. To enable large-scale phosphoproteomics studies with increased depth and throughput, it is necessary to reduce the amount of input protein, improve workflow reproducibility, and decrease mass spectrometry instrument time usage. With this in mind we optimized a fast and scalable single-shot analysis workflow based on high-throughput magnetic Ti-IMAC bead enrichment of phosphopeptides from 200 µg of starting tryptic peptide material (Fig. 1a). With this approach we routinely quantify ~7000 phosphopeptides in just 15 min of LC-MS/MS analysis time with the fast 28 Hz higher-energy collisional dissociation (HCD)[10] scanning method on a Q Exactive HF-X mass spectrometer[8] with overall MS/MS identification rates of more than 50%. This is close to 500 unique phosphopeptides per minute of gradient time, which is significantly more identifications than the commonly used TiO$_2$-based workflow[8].

However, even with the very fast and improved methodology, we seemed to have reached the limit for DDA for phosphoproteomics with current instrumentation. Conversely, DIA can in principle overcome this limitation of sequential DDA by analyzing peptide ions in parallel. This is achieved by co-isolating co-eluting peptide ions in predefined mass windows, fragmenting them together and analyzing all the resulting fragment ions simultaneously. However, few attempts have been reported on applying DIA to large-scale (phospho)proteomics[11–15]. An additional challenge in DIA is to localize phosphorylation sites correctly and handle positional phosphopeptide isomers[16]. To address this problem, we developed a PTM-specific workflow for peptide-centric DIA that combines the recovery rate of library-based extraction with high confidence site localization algorithm rivaling the current gold standard based on DDA.

Initially, we optimized the instrument settings for best DIA performance. Due to the co-fragmentation of multiple precursors with different charge states in DIA, it is not possible to employ the charge-state-dependent collision energy (CE) scaling of DDA. To identify the optimal CE settings for DIA, we recorded spectral libraries at four different normalized CE values and analyzed corresponding DIA runs with CE values fixed at charge-state of two. Analyzing the DIA files with the different spectral libraries revealed that the best compromise for maximizing identification of differently charged precursors was to record spectral libraries with NCE of 28 and analyze subsequent DIA runs with NCE of 25 at charge state 2 (Supplementary Fig. 1A). Next, we optimized the overlap between adjacent mass windows in DIA for best quantification by quantifying precursors in overlapping regions in a DIA setup in which we systematically shifted the mass windows. The best compromise between quantitative accuracy and number of identifications is reached with 1 Th overlap (Supplementary Fig. 1B). As an alternative strategy to identify the best overlap, we analyzed the relationship between ion transmission and centerness, where the transmission is defined as the relation between the intensity of the extreme $m/z$ compared to the intensity of the centermost precursor for DIA windows in two consecutive scan cycles shifted by half-a-window. The centerness is defined as the relationship between the extreme $m/z$ distance to center and ½ the window size. From this analysis, we also found that setting a fixed overlap of 1 Da between adjacent mass windows assured optimal quantification of precursors with $m/z$ values at the edges of isolation windows (Supplementary Fig. 1C). We also tested different DIA acquisition methods to find the optimal one for fast phosphoproteomics by changing scan cycle times using different mass window widths, number of windows, and HCD resolution settings (Supplementary Fig. 1D). All acquisition methods identified comparable numbers of phosphopeptides, which in DIA is defined as unique phosphorylated elution group precursors. However, the best quantitative performance judged by coefficient of variation (CV) between replica was achieved by the fastest scanning method employing 2 s cycle time with 48 mass windows of 14 Da widths using 15,000 resolution HCD

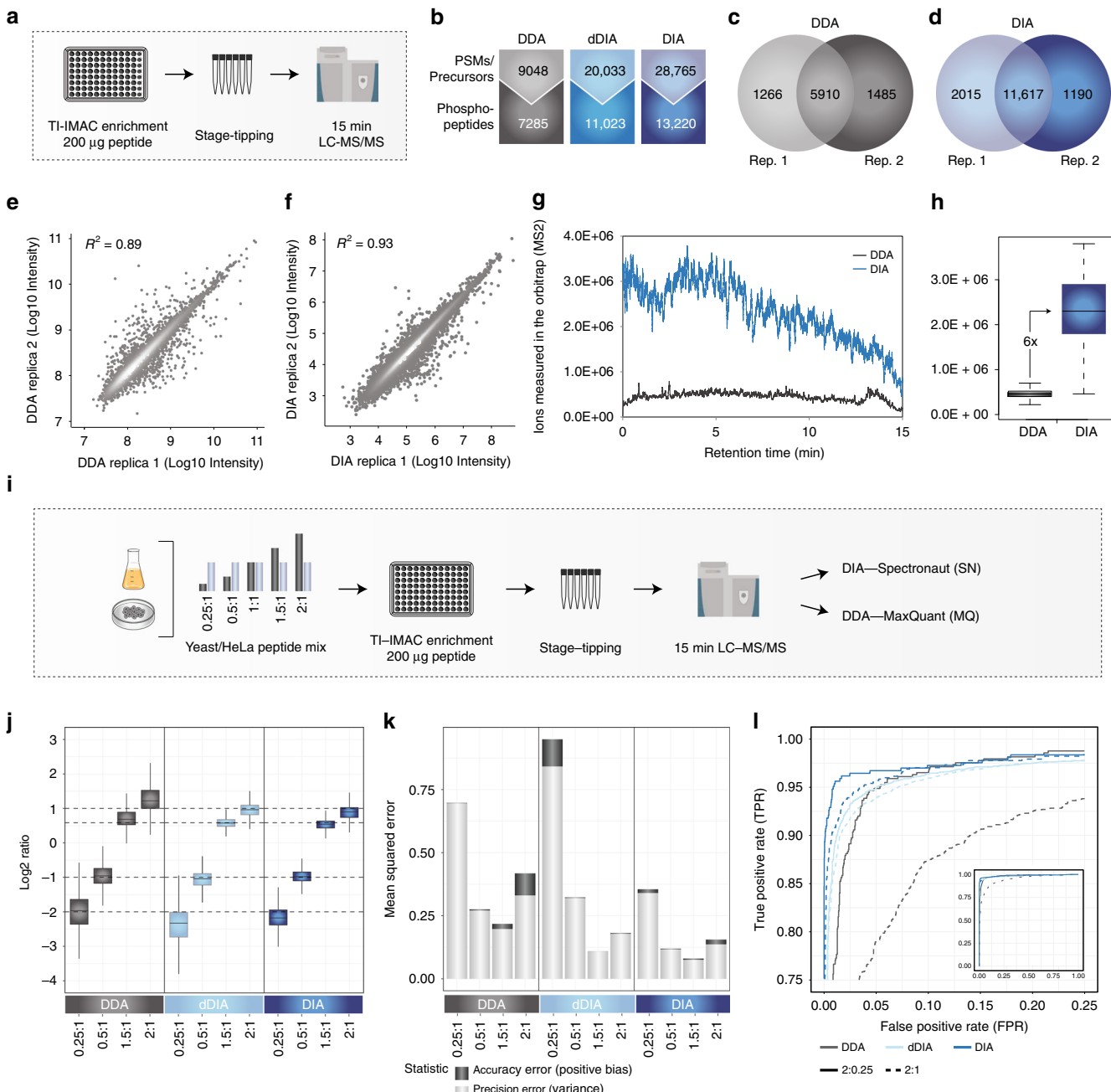

**Fig. 1 High-throughput and sensitive phosphoproteomics for DDA and DIA—identification and quantification. a** Experimental workflow for phosphoproteomics. **b** Comparison of quantified phosphopeptides with DDA & DIA. **c** Overlap of phosphopeptides between two replica with DDA. **d** Overlap of phosphopeptides between two replica with DIA. **e** Correlation between replicates with DDA. **f** Correlation between replicates with DIA. **g** Ions measured in the orbitrap in MS2 scans with DDA and DIA. **h** Quantified difference in ions measured in the orbitrap for DDA and DIA. **i** Experimental workflow for evaluation of accuracy and precision of the method. **j** Boxplot of measured and theoretical ratios for yeast phosphopeptides with DDA and DIA from six independent measurements. Boxes mark the first and third quantile, with the median highlighted as dash, and whiskers marking the minimum/maximum value within 1.5 interquartile range. Outliers are not shown. **k** Mean squared errors for DDA and DIA from six independent measurements were calculated as a sum of positive bias and variance for each method and all replicates. **l** Receiver operating characteristic (ROC) curves for DDA and DIA from six independent measurements were calculated by using the *d*-score from SAM testing as an indicator for significant regulation. SAM testing for significantly regulated phosphopeptides was performed at default settings (s0 estimation automatic). ROC plots are presented as zoomed-in excerpts from the total plots, shown on the lower right each. Source data for this figure are provided as a Source Data file.

fragmentation with maximum injection time of 22 ms (Supplementary Fig. 1E). Another important measure for evaluating quantitative accuracy of DIA acquisition methods is the number of data points measured across the elution profiles for precursors identified. Analyzing this for the different DIA acquisition methods reveals that the shortest DIA scan cycle time of 2 s

provides the highest number of data points per peak and hence best quantification (Supplementary Fig. 1F).

Using this optimized DIA method with 15 min LC gradients, we identified almost three times as many elution group precursors and twice as many phosphopeptides compared to the number of DDA peptide-spectrum matches and

phosphopeptides, respectively (Fig. 1b, see Methods section). In addition, the DIA raw files were also searched with direct DIA (dDIA). In this approach, spectral libraries are generated directly by searching deconvoluted pseudo-MS/MS spectra from DIA data against a peptide database, a strategy similar to DIA-Umpire[17]. For this process, Pulsar, the search engine in Spectronaut, applies the same search settings as DDA searches in MaxQuant. This library-independent dDIA strategy also worked well with twofold increase in precursors matched and 75% increase in phosphopeptides (Fig. 1b). DIA further showed a significantly higher overlap of phosphopeptide identifications between replica compared to DDA (Fig. 1c, d). Importantly, the quantitative reproducibility was better in DIA with correlation coefficient, $R^2$ of 0.93 compared to $R^2$ of 0.89 for DDA, even though DIA covered an additional order of magnitude of absolute precursor intensities (Fig. 1e, f). The observed increase in phosphopeptide identifications is likely due to the fact that DIA makes more efficient use of the ion beam by sampling more ions in MS/MS mode compared to DDA throughout the entire LC-MS analysis (Fig. 1g). Quantifying the difference reveals an approximate sixfold higher fragment ion count measured in DIA compared to DDA mode (Fig. 1h). This difference is to be expected as a target value of 3e6 was used for MS/MS scans in DIA experiments compared to a target value of 1e5 for DDA-MS/MS. Moreover, the higher reproducibility of DIA compared to DDA likely comes from the multiple fragments measured over an elution profile, whereas single elution profile quantification is performed in DDA.

We next assessed the quantification accuracy and precision of the DIA and DDA approaches in a regulated phosphopeptide sub-population spiked into a complex sample background. For this purpose, we made use of a mixed-species approach in which we diluted phosphopeptides enriched from yeast at different ratios into a fixed background of HeLa phosphopeptides and analyzed them by DDA, dDIA, and DIA (Fig. 1i, Supplementary Data 1). This strategy allows to assess how the acquisition methods quantify the expected ratios of the yeast phosphopeptides of 0.25:1, 0.5:1, 1.5:1, and 2:1. As expected, dDIA and DIA were both able to quantify up to twice as many phosphopeptides as DDA (Supplementary Fig. 1G). Based on boxplot analysis, all three methods very accurately estimated the expected ratios on median across all comparisons (Fig. 1j). From the boxplots it looks like DDA is slightly overestimating the ratios, whereas DIA is slightly underestimating the ratios. A likely explanation for this could be the difference in how the quantification is performed. DDA is based on full-scan MS1 quantification, where the preset target value is usually reached. Conversely, quantification in DIA is performed on the MS/MS level, where the preset target value is often not reached within the maximum allowed injection time for each DIA scan. However, the interquartile ranges were significantly smaller for DIA compared to DDA and generally dropped at the higher loads (2:1) compared to the more dilute sample (0.25:1). This indicates that phosphopeptides of higher intensity are better quantified as expected. Conversely, the human phosphopeptides did not show regulation between the conditions (Supplementary Fig. 1H). Likewise, an analysis of variance (ANOVA) statistical test revealed a higher number of significantly regulated yeast phosphopeptides for the DIA methods compared to DDA (Supplementary Fig. 1I).

To better assess quantification precision, we next calculated the mean squared error (MSE) as the sum of positive bias and variance for each method, which represent the quantification error in accuracy and precision, respectively (Fig. 1k). Based on this, DIA yields the highest precision and highest accuracy at all ratios analyzed. DDA showed lower precision compared to dDIA at high intensity ratios (2:1), but better precision at low intensity ratios (0.25:1). This is likely due to the way dDIA works as it is

based on DDA-like database search of the pseudo-MS/MS spectra derived from the DIA analysis. The DIA MS/MS spectra are generally much more complex than DDA-MS/MS, and therefore the identifications by dDIA pseudo-MS/MS rely more on fragment ions of higher abundance.

To test if the accurate and precise quantification of DIA translated into a better identification of significant regulation, we used the significance d-score of the SAM test[18] to calculate true-positive rates (TPR) and false-positive rates (FPR) of the regulated phosphopeptides for the DDA and DIA approaches (Fig. 1l). Plotting them against each other created a receiver operating characteristic (ROC) curve, in which best performing methods should achieve a TPR of 1 before increasing their FPR over 0. Focusing on the left part of the ROC curve plot, where the FPR is lowest, we see that DIA and dDIA shows the steepest TPR increase at all tested ratios compared to DDA.

**DIA-specific phosphorylation site localization algorithm**. To compensate for the wider isolation windows applied in DIA, we developed a PTM localization algorithm for peptide-centric analysis utilizing information not available in standard DDA data. This includes full isotopic patterns for fragment ions and the possibility of generating short elution chromatograms to correlate with the targeted precursor peak shape. The latter allows for systematic removal of any interfering fragment ions that one could not account for in DDA. These two aspects are combined with additional scores based on fragment ion intensities and mass accuracy into a specific weighted score for each fragment, which is then used to calculate a specific site localization score (Supplementary Note 1).

Briefly, during the standard DIA analysis, the algorithm starts out by detecting and classifying all potential peak groups for a peptide precursor in the library. For each candidate peak group modified peptides are enumerated into site candidates to represent all possible site combinations on-the-fly. Using the combined site information for a given peptide, the algorithm calculates and retains all unique fragment masses, which are then matched to the individual site candidates, whereby each individual fragment can be annotated as either confirming or refuting a specific site candidate (Fig. 2a). A score is calculated for each site candidate using the individual fragment ion matches incorporating aspects of the feature, the mass accuracy and the XIC correlation. The final site scores are then calculated by summing all candidate scores that supported this site and subtracting all fragments that refute the site (Fig. 2b). The developed DIA-specific PTM site localization score is finally calculated as a fractional site confidence score for each site combination compared in relation to all candidate scores. This approach is equivalent to the original PTM score site localization algorithm[2] and the Andromeda score[19] implemented in MaxQuant. The DIA-based PTM localization workflow does not require specially generated spectral libraries and is available in the Spectronaut software tool[7] (v. 13.0.190309.20491). An example of the four steps of the site localization algorithm for one doubly-phosphorylated peptide is demonstrated (Supplementary Data 2).

To evaluate the performance of the PTM site localization algorithm and establish appropriate site confidence score cutoff values for accurate site localization, we initially tested it on a library of synthetic phosphopeptides with known site localization. This library consisted of two hundred human tryptic phosphopeptides frequently observed in large-scale phosphoproteomics experiments. They were spiked into a stable background of tryptic yeast phosphoproteome sample in different concentrations (1×, 10×, 100×, 1000×) and measured in triplicates using DDA and DIA (Fig. 2c). The DDA raw files were analyzed with

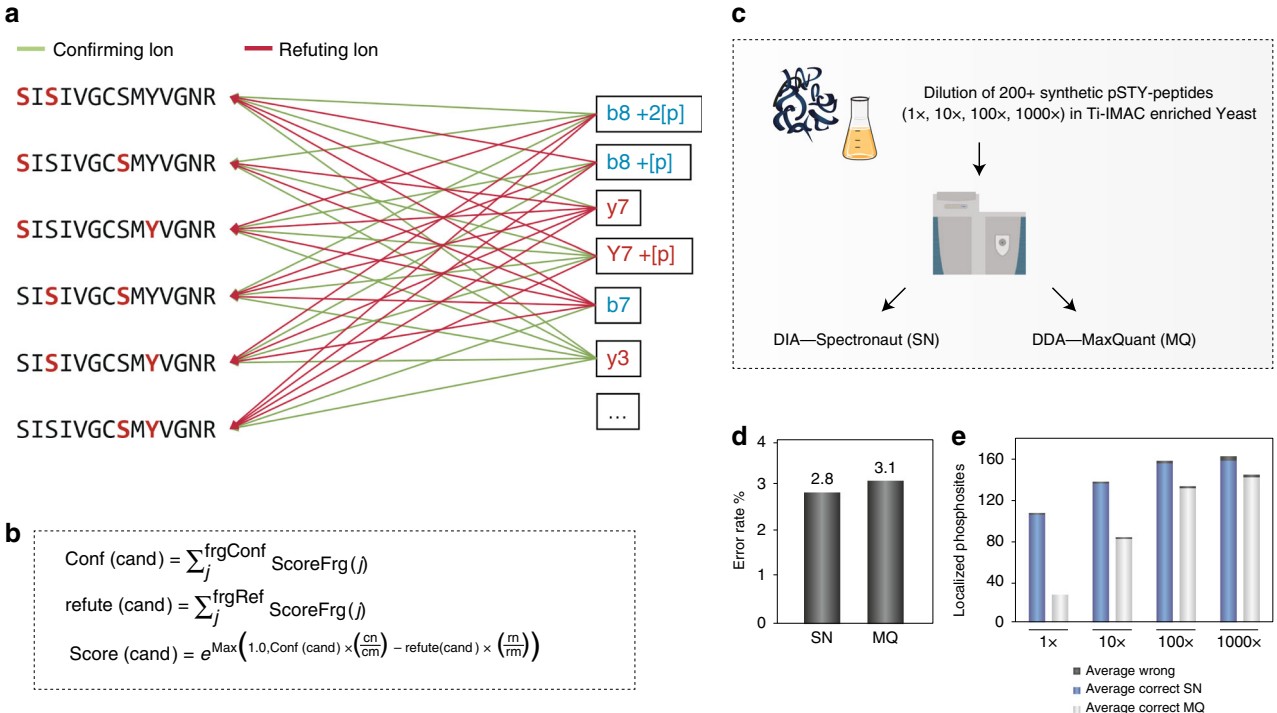

**Fig. 2 Challenges and solutions for phosphoDIA. a** Confirming and refuting fragments for site localization. **b** Calculation of site localization confidence score. The score for a given site localization candidate is calculated by summing all confirming fragments matched and subtracting the sum of all refuting fragments matched. **c** Workflow to evaluate localization algorithm with MaxQuant. **d** Error rates for phosphorylation site assignment in DDA and DIA. **e** Coverage of assigned phosphorylation site in a diluted yeast background for DDA and DIA. Source data for this figure are provided as a Source Data file.

MaxQuant[20] using phosphorylation (STY) as the only variable modification. The resulting phosphopeptide identifications were used to build a spectral library, which was then searched in peptide-centric mode against the corresponding DIA raw files using Spectronaut with the PTM site localization algorithm implemented.

From the DDA files, we on average correctly localized 108 phosphorylation sites from the synthetic phosphopeptides with 3.1% error rate on wrongly assigned sites (Fig. 2d). We required at least 0.75 localization site confidence (Class I sites) as in previous analyses[2]. Applying the same score cutoff of 0.75 for the DIA dataset results in correct identification and localization of 153.8 phosphorylation sites on average with 2.8% error rate of incorrectly assigned sites. This indicates that site localization FDR at this cutoff value is comparable to that of DDA analyzed with MaxQuant, but achieving higher site coverage in DIA. Notably, in DDA the number of synthetic phosphorylation sites identified is significantly hampered at low dilutions. Conversely, DIA maintained a relatively high identification rate across all dilutions, indicating that DIA outperformed DDA in sensitivity and dynamic range (Fig. 2e). We also analyzed the DIA dataset of the synthetic phosphopeptides using dDIA without a spectral library and found that in this case we needed to apply a higher site confidence score cutoff of 0.99 to achieve error rates comparable to library-based DIA and DDA (Supplementary Fig. 2A). However, even with this stringent cutoff, dDIA on average correctly identified and localized 143 phosphorylation sites, which was one-third more than DDA (Supplementary Fig. 2B).

**Technical comparison of DDA and DIA in a biological setting.** We next evaluated if the increased coverage of localized phosphopeptides in DIA over DDA translates into an advantage in a cell signaling study. For this purpose, we used EGF-stimulated retinal pigment epithelium (RPE1) cells treated with different MEK kinase inhibitors as a model system (Fig. 3a). Briefly, cells were pretreated for 30 min with 0.5 or 5 μM of Cobimetinib, or 0.5 or 5 μM of PD-032591 prior to 10 min EGF stimulation, 10 min EGF only or untreated cells as a control. All conditions were prepared as biological triplicates, phosphopeptides from 200 μg of whole-cell tryptic digests were enriched by Ti-IMAC and analyzed with 15-min gradients by DDA and DIA.

For DIA, we tested the effect of using different spectral libraries and dDIA. We recorded a project-specific spectral library consisting of >70,000 unique phosphopeptides by deep phosphoproteome profiling[21]. We fractionated the same EGF-stimulated cells using massive offline high-pH reversed-phase chromatography based fractionation combined with DDA analysis of individual fractions measured on the same 15 min online LC-MS gradient as for the DIA analyses, which should be an ideal reference dataset. As an alternative to this, we created an even larger community-based spectral library of >75,000 phosphopeptides by combining two previous large-scale HeLa (phospho) proteome studies[22,23]. About half of the phosphopeptides are present in both the project-specific library and the community-based library (Supplementary Figs. 3A, B).

To facilitate efficient bioinformatics analysis of phosphoproteomics data, MaxQuant generates a site-level output table for each variable PTM that allows site-level statistical analysis. To compare DIA Spectronaut data to DDA MaxQuant data on a biological level, we developed a Perseus plugin that can convert a normal Spectronaut report into a site-level report (Supplementary Note 2). The plugin features a graphical interface and allows generation of MaxQuant-like site-level, PTM-localized peptide-level, and modification-specific peptide-level output. With comparable data formats for both DDA and DIA, we first looked at the numbers of identified phosphopeptides and localized

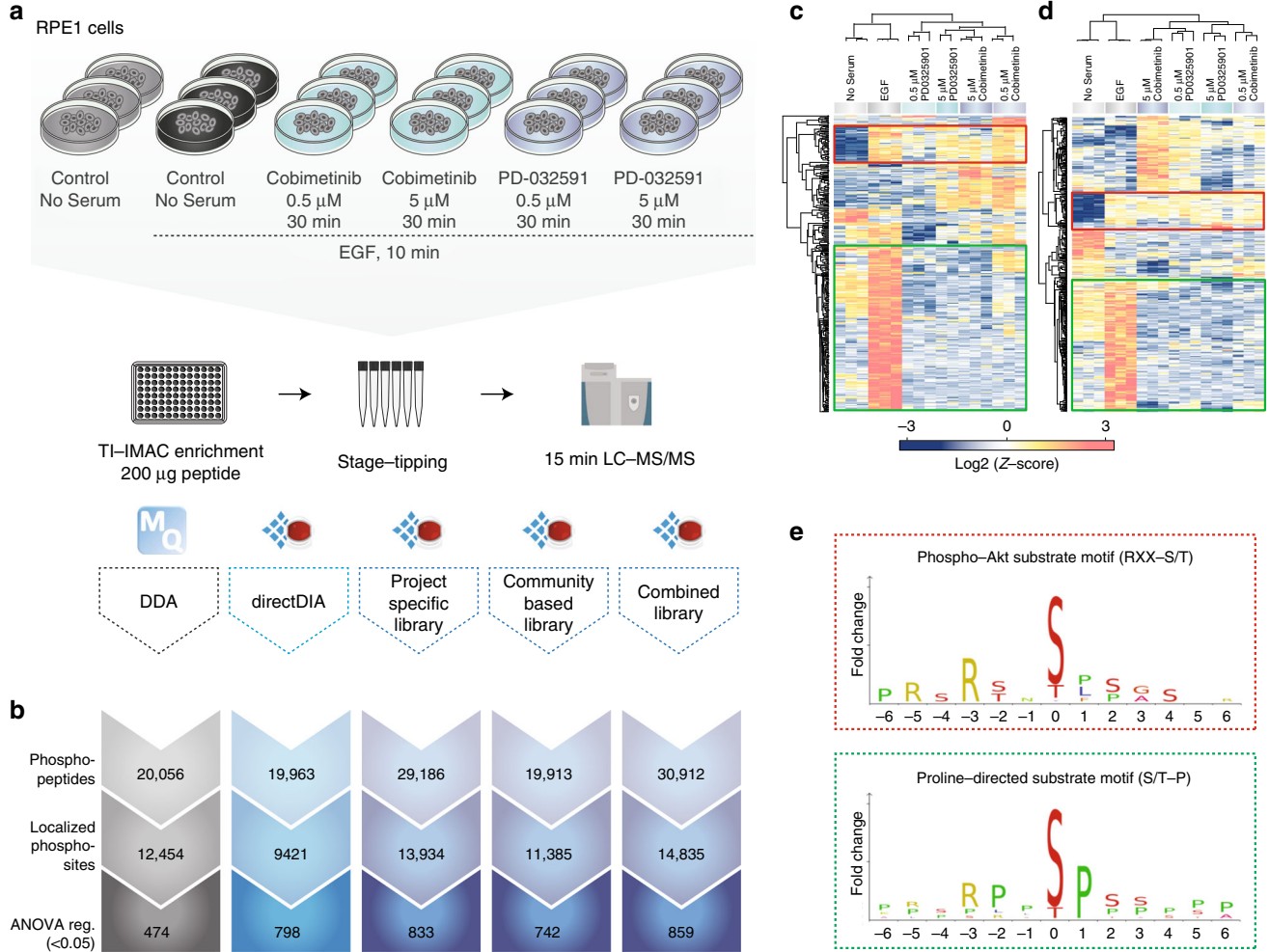

**Fig. 3 Technical comparison of DDA and different types of DIA in a biological setting. a** Experimental workflow. **b** Overview of identified phosphopeptides, localized phosphosites, and ANOVA ($s0 = 0.1$, FDR 0.5) regulated sites for the different methods. **c** Heatmap of unsupervised clustering analysis of ANOVA-regulated phosphosites for DDA workflow (**d**) and for DIA workflow with project-specific library (**e**). Linear sequence motif analysis for two major clusters marked in colored boxes on heatmaps. All data are shown for three independent measurements.

phosphorylation sites for each of the different methods, with more than 10,000 sites identified in all (Fig. 3b). As expected, DDA with 20,056 phosphopeptides covering 12,454 sites identified less sites than DIA with project-specific spectral library with 29,186 phosphopeptides covering 13,934 sites. Conversely, the larger community-based library only identified 19,913 phosphopeptides and 11,385 sites, which is comparable to DDA. However, combining the two spectral libraries provides the best coverage with 30,912 phosphopeptides and 14,835 sites (Supplementary Data 3). Interestingly, dDIA yields similar number of phosphopeptides and sites as DDA and DIA with the community library.

To assess the biological effect of the different kinase inhibitor treatments on cellular phospho-signaling, we performed an analysis of variance (ANOVA) statistical test to identify significantly regulated sites. For this test, we only used phosphorylation site ratios quantified in all three biological replicates of at least one condition. Due to its lower precision and phosphopeptide coverage, DDA yielded 474 significantly regulated phosphorylation sites, whereas all DIA methods identified nearly twice the number of regulated sites (Fig. 3b, Supplementary Data 3).

Next, we wanted to assess if and to what degree the regulated sites identified by DDA and DIA provided biological insights into EGF-dependent phospho-signaling in context of MEK inhibition

as expected. To do this, we performed unsupervised hierarchical clustering of the DDA significant sites (Fig. 3c) and the DIA significant sites (Fig. 3d), which revealed an overall similar pattern of regulation between the different conditions. The pattern was the same for the other DIA methods (Supplementary Fig. 3C). Linear sequence motif analysis of the EGF-upregulated phosphorylation sites that were largely unaffected by the kinase inhibitor treatment revealed that both DDA and DIA could correctly identify the EGF-dependent but MEK-independent AKT kinase substrate motif RxRxx[s/t]. Reassuringly, the ERK1/2 kinase substrate motif Px[s/t]P was significantly enriched among the MEK-dependent sites as expected (Fig. 3e). This analysis demonstrated that DIA and DDA identified the same biology on localized phosphosite-level—AKT and ERK activation as the major signaling axes downstream of EGF—recapitulating known EGF receptor signaling as expected.

Correct biological interpretation of phosphoproteomics data is dependent of the precision and accuracy of quantification for the identified phosphopeptides[24]. To evaluate the quality of label-free quantification methods for providing insights into EGF-signaling, we benchmarked our DIA and DDA datasets against a gold-standard reference dataset of EGF-dependent phosphorylation sites dynamics[2]. Of the 1050 dynamically regulated phosphorylation sites previously identified by SILAC-based quantitation of

five EGF stimulation time points of HeLa cells[2] (Supplementary Table 6), we covered 597 of them in our label-free DIA analysis and 504 of these sites in the DDA analysis, respectively. Note that we analyzed RPE1 cells in the current study and not HeLa cells, and perfect overlap is therefore not to be expected due to significant differences in protein abundances and hence phospho-signaling network activities between cell lines[23]. We only analyzed a single EGF stimulation time (10 min) in RPE1 cells and the fraction of regulated sites accounts for roughly 7% of all quantified sites, whereas many of the dynamic sites in HeLa cells were regulated at early (1–5 min) or late (20 min) EGF stimulation times. Of the overlapping EGF-dependent HeLa sites, 189 were ANOVA significant in the DIA analysis, whereas only 81 of them were ANOVA significant in the DDA analysis emphasizing the power of DIA for covering more relevant sites. As anticipated, the ANOVA significant DIA sites were mainly upregulated in HeLa cells after 10 min EGF stimulation with an average log2 fold-change of 1.0 and belonging to the clusters B, C, and D of intermediate stimulators, late stimulators, and terminal effectors defined in the HeLa dataset[2]. Conversely, the non-significant sites had an average log2 fold-change after 10 min EGF of 0.5, and they mainly belonged to clusters E and F of early negative regulators and late negative regulators with maximum phosphorylation site changes at 20 min of EGF stimulation. Performing a pathway-enrichment analysis of the 189 significant sites from the DIA dataset revealed the EGFR1 signaling pathway as the most significantly overrepresented pathway ($p = 1.57E-11$ and Benjami–Hochberg correct $p = 6.06E-09$) among the 746 distinct pathways covered (Supplementary Data 4). This benchmark underlines the high quality of DIA-based quantification for revealing biological insights.

**Analysis of fractional phosphorylation site stoichiometry.** Besides relative quantification of phosphorylation sites, it is valuable to determine their occupancy or absolute stoichiometry. A high fractional stoichiometry combined with dynamic regulation is a strong indication that the site is functional in the cellular context studied[1]. It is possible to determine the fractional stoichiometry of phosphorylation sites on a large scale by using ratios observed in both the phosphopeptide, its non-phosphorylated counterpart peptide and the respective protein between treatment conditions from SILAC data[25] and TMT-multiplexed data[24,26].

Based on our previous findings[24], we reasoned that the high quantitative accuracy and the completeness of the DIA phosphoproteomics dataset should allow the extraction of stoichiometry from multiple conditions at the same time. We adapted a recently developed 3D multiple regression model (3DMM)-based approach based on TMT data to label-free DIA. The 3DMM approach integrates information of several experimental conditions and replicates into one stoichiometry model, which uses phosphopeptide-, non-phosphorylated peptide- and corresponding protein-intensities from any multiplexed quantification method. However, even for DIA, data quantification of individual sites are not complete when analyzing many experimental conditions. To overcome this issue and retain as much quantitative information as possible, we combined peptide information based on the assumption of linear behavior between equally (non-)regulated peptides, which allowed us to extrapolate peptide intensities to fill in missing values. We implemented both the linear modeling approach and stoichiometry calculation into our Perseus plugin, which thus allows users without prior scripting experience to calculate PTM occupancies from LFQ data (Supplementary Note 2).

To benchmark the performance of the label-free 3DMM stoichiometry approach, we prepared a mixed-species sample with fixed phosphopeptide stoichiometries (Fig. 4a).

A phosphopeptide-enriched tryptic yeast digest was split in two equal parts and one half was dephosphorylated using alkaline phosphatase. Mixing together phosphorylated and non-phosphorylated yeast peptides in fixed ratios into a HeLa phosphopeptide background yielded conditions of 1%, 10%, 50%, 90%, and 99% phosphorylation site stoichiometry. Each sample was analyzed by DDA and DIA using 15 min LC-MS/MS (Supplementary Data 5). Boxplot analysis of the calculated stoichiometry values revealed that both DDA and DIA estimated the expected ratios closely on median across all comparisons.

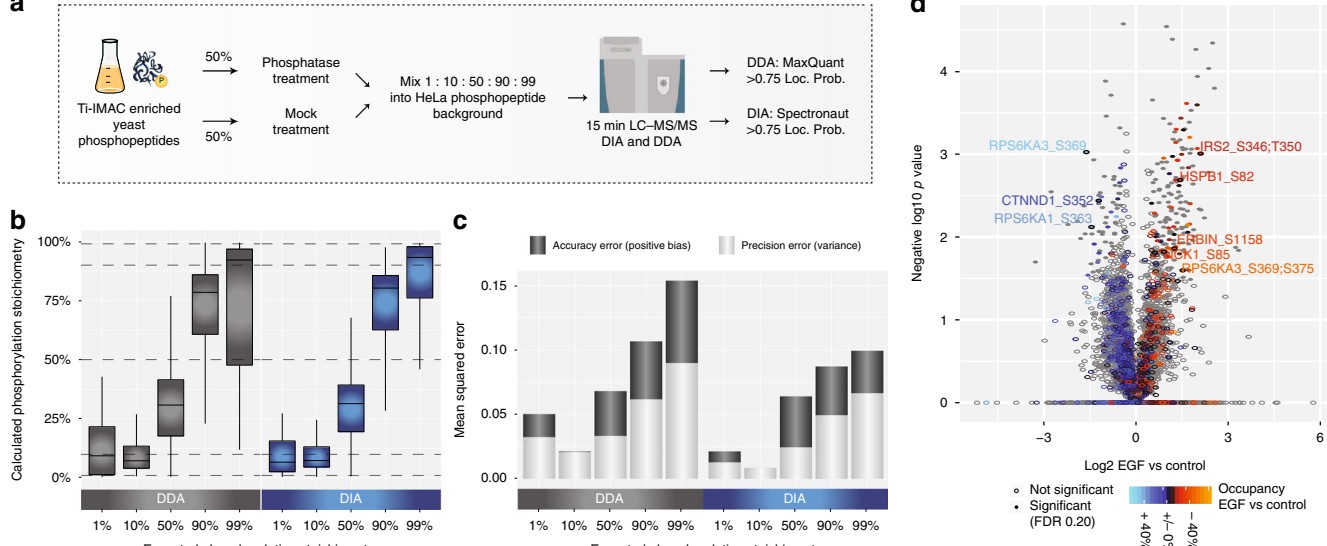

**Fig. 4 Stoichiometry benchmark. a** Experimental workflow for experiment with controlled ratios. **b** Boxplot of calculated phosphorylation stoichiometry with DDA and DIA from six independent measurements. Boxes mark the first and third quantile, with the median highlighted as dash, and whiskers marking the minimum/maximum value within 1.5 interquartile range. Outliers are not shown. **c** Mean squared errors for calculated stoichiometries with DDA and DIA from six independent measurements were calculated as a sum of positive bias and variance for each method and all replicates. **d** Volcano plot analysis of calculated occupancies EGF vs. control from three independent measurements. Source data for this figure are provided as a Source Data file.

However, the interquartile ranges were significantly smaller for DIA compared to DDA especially at extreme stoichiometry values, indicating that higher intensity phosphopeptides are better quantified by DIA as expected (Fig. 4b). To compare the quantification error in terms of accuracy and precision for the 3DMM data, we compared the mean squared errors (MSE) presenting the sum of positive bias and variance for the two methods, respectively, as described above. Based on this, DIA yields the highest precision and highest accuracy at all stoichiometry levels analyzed with the general trend that precision is lowest for highest values, whereas accuracy is more comparable across conditions (Fig. 4c). To demonstrate that DIA and DDA correctly estimate the expected ratios, we calculated mean standard deviations and median absolute deviations for each stoichiometry mix (Supplementary Fig. 4A). These values were calculated for all stoichiometries across all replicates per condition. Together with the calculated CVs for the different dilutions it is clear that DIA outperforms DDA in all dilutions (Supplementary Fig. 4B).

Our data indicate that DIA-based stoichiometry estimation is possible with reasonable accuracy and we therefore applied it to the EGF-stimulated and kinase-inhibitor-treated RPE1 cell phosphoproteome DIA dataset described above. Heatmap representation of the unsupervised hierarchical clustering of ANOVA significant phosphorylation site stoichiometries reflected the cellular conditions well with generally highest stoichiometry in EGF-stimulated samples and lowest in kinase inhibitor treated (Supplementary Fig. 4C). This set of dynamic phosphorylation sites with high stoichiometry values were enriched in proteins associated with receptor tyrosine kinase (RTK) signaling according to the Reactome Pathway Database[27]. To pinpoint likely functional sites to prioritize for follow-up experiments, the global phosphorylation site stoichiometry measurements can be integrated with the corresponding site fold-changes and visualized as an extra layer of information in a Volcano plot. This is exemplified by the comparison of EGF versus control samples using the phosphoproteomics dataset from Fig. 3 based on the project-specific library (Fig. 4d). Enrichment analysis among the significantly EGF-regulated site occupancies reveal strong overrepresentation of signaling by receptor tyrosine kinases and MAPK signaling pathways validating the known biology of the experiments (Supplementary Fig. 4D).

**Large-scale phosphoproteomics using a kinase inhibitor panel.** To demonstrate the power and scalability of the rapid DIA-based site-specific phosphoproteomics workflow developed here, we applied it to identify phosphorylation site targets of the ten major protein kinases in the epidermal growth factor signaling pathway using a panel of 30 kinase inhibitors (Fig. 5a).

Briefly, EGF-stimulated RPE1 cells were pretreated with an inhibitor in two different concentrations (0.1 and 1 μM) in biological triplicates and each of the 186 samples was analyzed by 15 min LC-MS/MS using DIA. We quantified ~20,000 phosphopeptides across the 62 conditions in triplicates and performed ANOVA significance analysis on the log-transformed normalized intensities, which identified 1275 phosphorylation sites that were regulated in at least one condition (Supplementary Data 6). We visualized the regulated sites as a function of treatment by hierarchical clustering of the averaged phosphorylation site intensities per replica, which grouped likely substrates and targets according to the kinase inhibited (Fig. 5b). From this analysis it is evident that EGF receptor inhibition (EGFRi) by all compounds worked well as it clusters with the untreated control samples. To verify that the inhibitors targeted the expected kinases, we performed a kinase motif enrichment analysis among each of the

down-regulated phosphorylation site clusters for the individual kinase classes using a Fisher exact test. The overrepresented kinase motifs generally matched the expected kinase or their main established downstream kinase substrates (Fig. 5c). For example, MEK inhibition leads to strong overrepresentation of ERK1/2 motif as expected, GSK3 inhibition down-regulates sites that conform to the known GSK3 motif, and mTOR inhibition down-regulated sites with the known mTOR substrate motif as well as substrate sites of its main downstream kinase p70S6K. To further validate the specificity of the kinase inhibitors we analyzed known substrates of the individual kinases and found good reproducibility in our dataset (Fig. 5d, Supplementary Data 7).

## Discussion

Here we optimized a streamlined phosphoproteomics workflow based on DIA and developed a PTM localization site algorithm as part of the DIA computational pipeline, which we benchmarked against state-of-the-art DDA-based phosphoproteomics. By analyzing a library of synthetic phosphopeptides, controlled mixed-species phosphoproteomes with technical replication and EGF-stimulated cells in combination with kinase inhibitors with biological replication, we show the robustness, specificity, and high quality of the DIA-based phosphoproteomes. Quantitatively we demonstrate that we can achieve significant greater depth than any previous DIA-based phosphoproteome reported. Moreover, we present a large-scale systematic analysis of the effects of kinase inhibitors on phosphoproteomes using DIA-based label-free quantification.

More generally, the methodological approach we have developed represents a strategy to quantitatively profile hundreds of phopshoproteomes in a few days from a low amount of starting material. While it has been shown previously that rapid plate-based enrichment for DDA-based sensitive phosphoproteomics starting from 200 μg or less[28,29] is feasible, this study represents an important advancement in that we use sensitive phosphoproteomics to analyze cellular signaling networks much faster and with greater depth. Furthermore, we demonstrate that stoichiometry calculation using LFQ is principally feasible, which has the potential to help in identifying functionally relevant phosphorylation sites in the future.

There are still limitations to the optimal DIA-based phosphoproteomics workflow. We achieve the best coverage and quantification when using tailor-made project-specific spectral library for the DIA analyses, but this requires some effort and may not always be possible. However, predicting peptide retention time and MS/MS spectra[30,31] might circumvent the necessity of recording spectral libraries for DIA in future. Unfortunately, the current prediction tools are not yet developed for phosphoproteomics.

Alternatively, library-free approaches such as dDIA look very promising for the future. Although classic DIA analysis with an experiment specific spectral library is superior to dDIA, it is much easier to implement the dDIA workflow making it easier accessible to the general proteomics community. Importantly, dDIA also may overcome issues with standard DIA, where rare or low abundant phosphorylation sites may get diluted out during library generation. In a classic DIA analysis, a phosphorylation site must be present in the library for it to be considered. Conversely, in dDIA all possible phosphorylation site combinatorics for a given peptide is considered similar to DDA. Moreover, we believe that there is still room for improvements in the mass spectrometric technology. Although we show that DIA analyzes about sixfold more ions in MS/MS mode than DDA, we estimate that by using 48 DIA windows we are still maximally sampling a few percent of the ion beam at best. Mass spectrometers with higher duty cycles such as the timsTOF pro[32] may be an even better fit for DIA-based phosphoproteomics in the

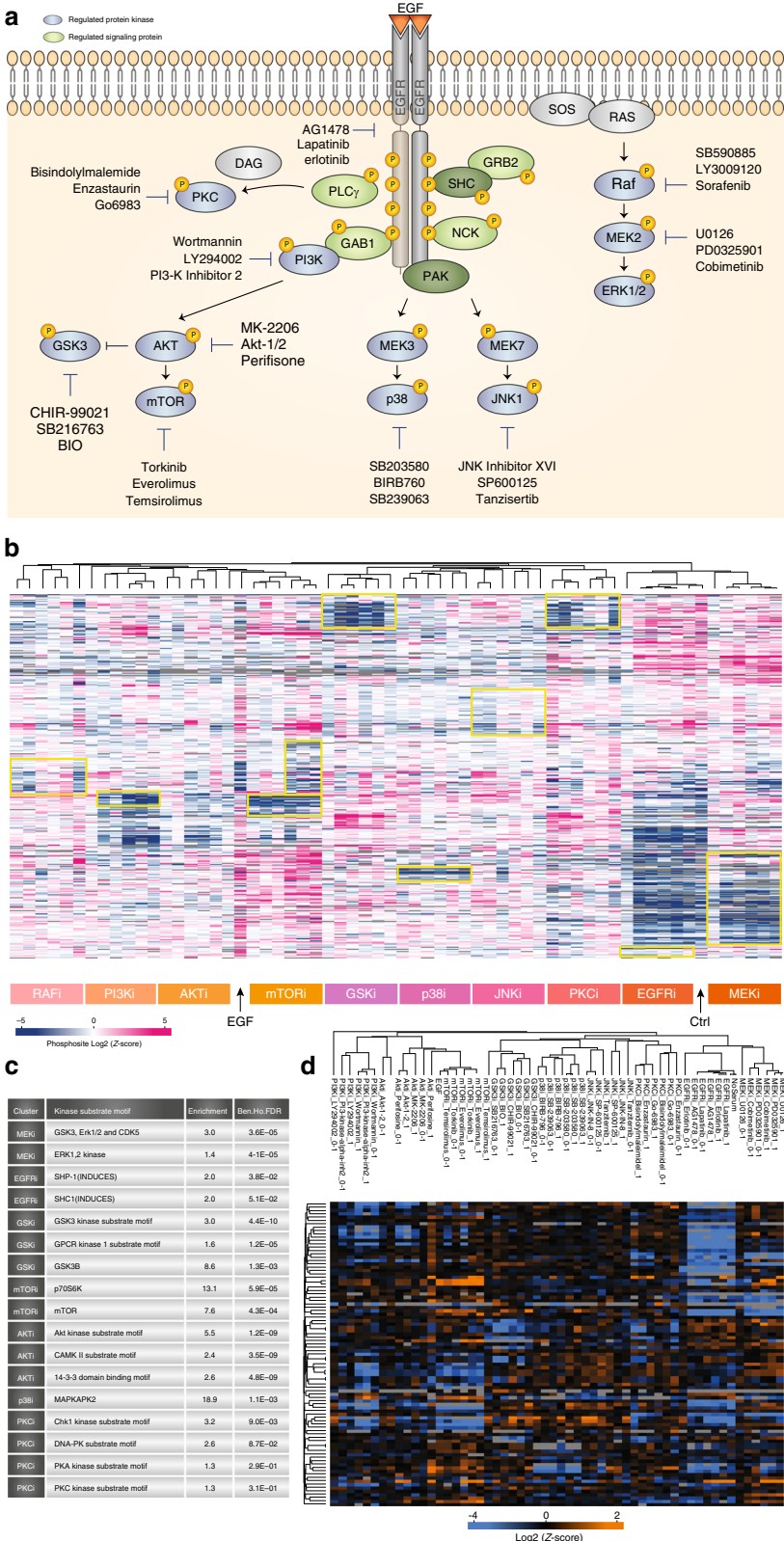

**Fig. 5 Kinase inhibitor screen. a** Experimental overview of kinase inhibitors used. **b** Hierarchical clustering of averaged site intensities from six independent measurements. **c** Fisher Exact test of overrepresented kinase motifs. **d** Clustering analysis of known substrates and individual kinases.

future. Looking forward, the methodological and computational framework outlined here may be applicable to a clinical setting for example in cancer, where sensitive phosphoproteomics profiling of individual patient tumors may aid precision medicine in the future.

## Methods

**Human cell culture and lysis**. Human epithelial cervix carcinoma HeLa cells (female) and human retinal pigment epithelial RPE1 cells (female) immortalized with hTERT were purchased from ATCC (CCL-2 and CRL-4002, respectively) and cultured in DMEM (Gibco, Invitrogen) supplemented with 10% fetal bovine serum

(Gibco, Invitrogen), 100 U/ml penicillin (Gibco, Invitrogen), 100 µg/ml streptomycin (Gibco, Invitrogen) at 37 °C in a humidified incubator with 5% $CO_2$. Cells were harvested at approximately 80% confluency by washing twice with PBS (Gibco, Invitrogen) and subsequently adding boiling GdmCl lysis buffer (6 M guanidine hydrochloride, 5 mM tris(2-carboxyethyl)phosphine, 10 mM chloroacetamide, 100 mM Tris pH 8.5) directly to the plate. Cells were collected by scraping the plate and boiled for 10 min at 95 °C followed by micro tip sonication.

**Yeast cell culture and lysis.** BY4742 wild-type cells (ThermoFisher Scientific) were grown in YEPD medium (2% bacto peptone, 1% yeast extract, 2% dextrose/glucose; sterile-filtered before use) at 30 °C and 200 r.p.m. rotation in overnight culture. Day culture was inoculated at OD_600 of 0.1 and harvest hours later when the OD_600 surpassed ~0.8. Yeast cells were spun down (4000g 5 min) and washed with ice cold PBS. The pellet was resuspended in yeast lysis buffer (20 ml per 1 liter OD_600 1; 75 mM Tris pH 8, 75 mM NaCl, 1 mM EDTA, 1 complete miniprotease inhibitor cocktail tablet per 10 ml, 5 mM sodium fluoride, 1 mM sodium orthovanadate, 5 mM β-glycerol phosphate) and dropped in droplets out of pipette into liquid nitrogen. Frozen droplets were ground in a MM400 ball mill (Retsch) for 3 min at 25 Hz. Frozen yeast powder was then mixed with 1% Triton X-100 and 0.5% SDS and incubated rolling at 4 °C until thawed. Yeast lysate was spun down (16,000 r.p.m. 4 °C 5 min). The supernatant was transferred into −80 °C acetone to a final acetone concentration of 80% v/v and incubated at −20 °C for 2 h. Precipitated proteins were spun down and resuspended in GdmCl lysis buffer (6 M guanidine hydrochloride, 5 mM tris(2-carboxyethyl)phosphine, 10 mM chloroacetamide, 100 mM Tris pH 8.5, 1 complete mini protease inhibitor cocktail tablet per 10 ml, 5 mM sodium fluoride, 1 mM sodium orthovanadate, 5 mM β-glycerol phosphate). Protein pellets were resuspended by sonication (Sonics & Materials, VCX 130; 1 s on, 1 s off, 80% amplitude) and boiled for 10 min at 95 °C.

**Protein digestion.** Protein concentration was estimated by BCA assay (Pierce) and the lysates were digested with Lys-C (Wako) in an enzyme/protein ratio of 1:100 (w/w) for 1 h followed by a threefold dilution with 25 mM Tris, pH 8.5 to 2 M GdmCl and further digested overnight with trypsin (Sigma-Aldrich) 1:100 (w:w). Protease activity was quenched by acidification with trifluoroacetic acid (TFA) to a final concentration of approximately 1% and the resulting peptide mixture was concentrated using reversed-phase Sep-Pak C18 cartridge (Waters). Peptides were eluted off the Sep-Pak with 2 ml 40% acetonitrile (ACN) followed by 2 ml 60% ACN. The ACN was removed by vacuum centrifugation at 60 °C and the final concentration was estimated by measuring absorbance at 280 nm on a NanoDrop 2000C (Thermo Scientific).

**Phosphopeptide enrichment.** Two hundred micrograms of peptides were enriched for phosphopeptides using Ti-IMAC magnetic beads (Resyn Biosciences). Enrichments were carried out in protein LoBind 96-well plates (Eppendorf). The plates were mixed using 1300 rpm (Heidolph Titramax 1000, #544-12200-00) and separated in the 96-well plate using a magnetic stand (Thermo Scientific, # AM10027). Ti-IMAC beads were equilibrated twice in 200 µl 70% ethanol followed by once in 100 µl 1% ammonia and three times in loading buffer (80% acetonitrile, 1 M glycolic acid, 5% TFA). Two hundred micrograpms peptide mixture was mixed with the equal amount of loading buffer and 500 µg Ti-IMAC beads (25 µl) were added and the solution was mixed for 20 min. The beads were separated on a magnetic stand for 20 s and the supernatant was removed using gel-loader tip connected to vacuum. Two hundred microliters of loading buffer were added and beads were mixed for 2 min followed by two washes with wash buffer 1 (80% acetonitrile, 1% TFA) and once with wash buffer 2 (10% acetonitrile, 0.2% TFA). Phosphopeptides were eluted in three rounds with 80 µl 1% ammonia for 20 min, transfered the supernatant to a clean plate, and acidified with TFA. Speedvac solution was desalted using C18 StageTips and stored at 4 °C until MS analysis.

**Preparation of spectral libraries.** RPE1 cells were stimulated with 125 ng/ml EGF (Chromotek) for 3 and 10 min followed by lysis and digestion as described before. Ultimate 3000 ultra-high performance liguid chromatography (UHPLC) (Dionex, Sunnyvale, CA, USA), in conjunction with high-pH reversed-phase chromatography, was used to separate and fractionate tryptic peptides. Peptides were separated using a high-pH-compatible 250 3 4.6 mm C18 Waters BEH X-Bridge peptide separation technology (PST) 3.6 mM or Phenomenex Cell Reports 22, 2784–2796, March 6, 2018 2793 Kinetex Evo 2.6 mM (Torrance, CA USA) column with identical dimensions. Basic conditions were achieved by running buffer C (50 mM ammonium hydroxide) constantly at 10% (100 ml/min, 2.5 mM ammonium bicarbonate final). A 60 min fractionation and collection gradient were achieved using buffer A (Milli-Q H2O) and buffer B (acetonitrile), and fractions were collected. From 3 mg, 10 fractions were collected, while 10 mg was fractionated and collected in 46 fractions without any concatenation. Running at a constant 1 ml/min, the gradient was increased from 5% to 25% buffer B in 50 min and further increased to 70% buffer B in 5 min, where it was held for another 5 min. At this point, the fraction collection was stopped. Each fraction was dried in a speedvac and reconstituted in phopsho-loading buffer and phosphopeptides were enriched as described above. Each fraction was analyzed individually with LC-MS/MS settings as described below.

**Stoichiometry.** Phosphopeptides enriched from yeast and HeLa cells as described above were each split 50:50. One-half was dephosphorylated using rAPid alkaline phosphatase (Sigma-Aldrich; 1 µl per 2 mg protein starting material) at 37 °C 750 r.p.m. shaking overnight, while the other half was mock treated using water. Both samples were incubated for 10 min at 85 °C to inactivate the phosphatase, and then purified on C18 StageTips[33]. Yeast phosphopeptides were mixed at 0.1%, 1%, 10%, 50%, 90%, 99%, and 99.9% into dephosphorylated peptides at 99.9%, 99%, 90%, 50%, 10%, 1%, and 0.1%, with the total amount of peptides per condition corresponding to Ti-IMAC enrichment from 200 µg yeast starting material. Mixtures were then added into a 1:1 HeLa mixture of phosphopeptides and dephosphorylated peptides, each corresponding to a total amount of peptides per condition corresponding to Ti-IMAC enrichment from 100 µg HeLa starting material. Samples were then measured with the DDA and optimized DIA method as described below.

**Synthetic phosphopeptide experiment.** Synthetic phosphopeptides were purchased from JPT (SpikeMix PTM-kit 52 1001098; SpikeMix PTM-kit 54 1001100) and Sigma-Aldrich (MS PhosphoMix 1 Light MSP1L, MS PhosphoMix 2 Light MSP2L, MS PhosphoMix 3 Light MSP3L) and resuspended according to the manufacturer's instructions. Synthetic phosphopeptides and yeast phosphopeptides as described above were mixed in different ratios to mimic complex phosphopeptide mixtures (Supplementary Data 1). Samples were measured in DIA and DDA mode in technical triplicates each. For DIA measurement, the optimized method described below with the LC-gradient scaled to 35 min was used. For DDA measurement, the same LC-gradient was measured with MS1 resolution 60,000, MS1 AGC target 3e6, MS1 max IT 45 ms, scan range 350–1400 $m/z$, MS2 resolution 30,000, MS2 AGC target 1e5, MS2 max IT 54 ms, MS2 top6, MS2 isolation window 1.3 $m/z$, MS2 scan range 200-2000 $m/z$, MS2 NCE 28%, and MS2 dynamic exclusion 30 s. DIA data were searched using a spectral library generated from the DDA files, searched with MaxQuant, and generated by Spectronaut as described below.

**Nanoflow LC-MS/MS.** The peptides were concentrated in a speedvac and volume were adjusted to 7 µl in loading buffer (5% ACN and 0.1% TFA) prior to auto-sampling. An in-house packed 15 cm, 75 µm ID capillary column with 1.9 µm Reprosil-Pur C18 beads (Dr. Maisch, Ammerbuch, Germany) was used. An EASY-nLC 1200 system (ThermoFisher Scientific, San Jose, CA) was used and the column temperature was maintained at 40 °C using an integrated column oven (PRSO-V1, Sonation, Biberach, Germany) interfaced online with the mass spectrometer. Formic acid (FA) 0.1% was used to buffer the pH in the two running buffers used. The total gradient time was 19 min and went from 8% to 24% acetonitrile (ACN) in 12.5 min, followed by 2.5 min to 36%. This was followed by a washout by a 1/2 min increase to 64% ACN, which was kept for 3.5 min. Flow rate was kept at 350 nL/min. Re-equilibration was done in parallel with sample pickup and prior to loading with a minimum requirement of 1 µl of 0.1% FA buffer at a pressure of 800 bar.

Spray voltage was set to 2 kV, funnel RF level at 40, and heated capillary at 275 °C. For DDA experiments full MS resolutions were set to 60,000 at $m/z$ 200 and full MS AGC target was 3E6 with an IT of 25 ms. Mass range was set to 350–1400. AGC target value for fragment spectra was set at 1E5 with a resolution of 15,000 and injection times of 22 ms and Top12. Intensity threshold was kept at 2E5. Isolation width was set at 1.3 $m/z$ and a fixed first mass of 100 $m/z$ was used. Normalized CE was set at 28%. For DIA experiments full MS resolutions were set to 120,000 at $m/z$ 200 and full MS AGC target was 3E6 with an IT of 45 ms. Mass range was set to 350–1400. AGC target value for fragment spectra was set at 3E6. In all, 48 windows of 14 Da were used with an overlap of 1 Da. Resolution was set to 15,000 and IT to 22 ms. Normalized CE was set at 25%. All data were acquired in profile mode using positive polarity and peptide match was set to off, and isotope exclusion was on.

**Raw data processing.** DDA files were processed using MaxQuant (1.6.5.0) with default settings. Carbamidomethyl (C) was set as fixed modifications. Oxidation (M), Acetyl (Protein N-term), Phospho (STY), Deamidation (NQ) and Gln->pyro-Glu were set as variable modifications. Reference FASTA files for human and *S. cerevisiae* were downloaded from Uniprot on 15th of March 2018.

Spectral libraries were built from MaxQuant DDA search results using Spectronaut Professional + x64 (13.0.190309.20491)[7] with default settings, but Best N Fragments per Peptide Max set to 25 instead of 6. The yeast phosphopeptide library was generated from yeast phosphopeptides fractionated into 12 and 46 fractions. We used an Ultimate 3000 HPLC system (Dionex) with a Waters Acquity CSH C18 1.7 µm 1 × 150 mm column on operating at a flow rate of 30 µl/min with two buffer lines as previously described[23].

DIA files were processed using Spectronaut with default settings, with PTM localization activated and site confidence score cutoff set to 0 (for peptide-level analysis of dilution benchmark), 0.75 (for spectral library analysis) or 0.99 (for direct DIA analysis), data filtering set to Q-value and Normalization Strategy set to Local Normalization. Unless otherwise stated, experiments containing yeast peptides were searched using the yeast phosphopeptide library, those containing human peptides using the human phosphopeptide library, and those containing both with both. For the DIA stoichiometry benchmark (Fig. 4a–c), DIA files were additionally searched with a library generated from the DDA runs of the same

experiment in order to have non-phosphopeptides represented in the library as well. The synthetic phosphopeptide benchmark (Fig. 2) was searched with a library generated from the DDA runs of the same experiment.

Direct DIA search was performed in Spectronaut using default settings with the same PTMs as defined in MaxQuant DDA searches, Data filtering set to Q-value and Normalization Strategy set to Local Normalization.

Transformation of the Spectronaut normal report and calculation of stoichiometry values was performed using a custom coded plugin Peptide Collapse in Perseus (1.6.5.0). The plugin was created using Microsoft Visual Studio 2017 (15.6.3) and requires Perseus and R (minimum version 3.6.0) to run. Detailed information on how to install and use the plugin and the calculations it performs are listed in Supplementary Note 2.

**dDIA analysis**. To search DIA data directly against a protein database FASTA file, the DIA raw dataset is first converted into DDA-like data of individual precursors and their associated fragment peak lists[17]. To generate such pseudo-DDA data, an MS1 feature detection is performed in three dimensions ($m/z$, intensity, and retention time). The resulting 3D features are created by connecting scan-wise 2D features. Finally, precursor-fragment groups are created by prioritizing fragment ions corresponding to the individual 3D precursor features. The pseudo-DDA data are then searched using the Pulsar search algorithm, which is a classical peptide search engine similar to Andromeda in MaxQuant and Mascot[19,34]. This approach allows searching complex DIA data in a similar fashion as classical DDA data without the need of a spectral library.

**Bioinformatics analysis**. Most of the data analysis was performed using custom scripts in R (64 bit version 3.6.0) with packages data.table (1.12.2), bit64 (0.9-7), doParallel (1.0.14), stringr (1.4.0), ggplot2 (3.1.1), qplot (3.0.1.1), limma (3.40.0), samr (3.0), magrittr (1.5), scales (1.0.0), XML (3.98-1.19), PerseusR (0.3.4), Biostrings (2.52.0), and MASS (7.3-51.4). ANOVA testing was performed in Perseus (1.6.5.0) and KEGG/reactome term enrichment was performed using the STRING app (1.4.2) in Cytoscape (3.7.0).

For the dilution benchmark experiment (Fig. 1i–l), Spectronaut reports were transformed into modification specific peptide-like reports using the plugin peptide collapse in Perseus, with EG.PTMAssayProbability as grouping column, localization cutoff 0 and same variable PTMs as listed above and summed intensities are log2-transformed. For the stoichiometry benchmark experiment (Fig. 4a–c) and the kinase inhibitor stoichiometry experiment (Fig. 4d), stoichiometry values on target PTM peptide level were calculated for both DDA and DIA data using the plugin peptide collapse in Perseus, with grouping columns Modified sequence and EG.PTMLocalizationProbabilities, respectively. Localization cutoff was set to 0.75 for both and same variable PTMs as listed above.

For the analysis of biological benchmark data and the kinase inhibitor screen, the Spectronaut-reported normalized intensity data matrices were transformed into phosphorylation site tables with the plugin peptide collapse script using localization cutoff of 0.75. Downstream data analysis was performed by filtering for minimum of three valid values in at least one treatment group followed by median subtraction across conditions. Missing values were imputed by random sampling of the lower end of the normal distribution (width 0.3 and downshift 1.8) in total matrix. ANOVA significance testing was performed on z-scored log-transformed phosphorylation site intensities with parameter settings of $s0 = 0.1$ and $FDR = 0.05$ or 0.01. Heatmaps of all significantly regulated phosphorylation sites were generated using unsupervised hierarchical clustering using Pearson correlation analysis.

For the LFQ dilution and stoichiometry benchmark experiment, spectronaut intensities were de-normalized by dividing reported intensity values by their normalization factors. For the stoichiometry benchmark experiment, DDA and DIA intensities were then quantile-normalized.

Boxplots were created with boxes marking the first and third quartile, a dash the median, and whiskers the minimum/maximum value within 1.5 interquartile range. Outliers are not displayed.

SAM testing was performed using default settings ($s0 = 0.1$, $FDR = 0.20$). ANOVA testing of stoichiometry values was performed using $s0 = 0.01$ and $FDR = 0.20$. Enrichment of KEGG/reactome terms was performed in the Cytoscape STRING app using default settings.

**Reporting summary**. Further information on research design is available in the Nature Research Reporting Summary linked to this article.

## Data availability

The mass spectrometry proteomics data have been deposited to the ProteomeXchange Consortium via the PRIDE partner repository. Data are available via ProteomeXchange with identifier PXD014525. A roadmap linking files in the ProteomeXchange folder to the figures in the manuscript is available (Supplementary Data 8). The source data underlining Figs. 1e–1h, 1j–1l, 2d–2e, 4b–4d and Supplementary Figs. 1e, 1g–1i, 2c–2d and 4b–4d are provided as a Source Data File. Data used for Figs. 3b–3d, 5b are 5d available as Supplementary Data 3, 6 and 7, respectively. All other data are available from the corresponding author on reasonable request.

## Code availability

Custom R code for the data analysis is available upon request. The Perseus plugin Peptide Collapse. The plugin requires Perseus and R (minimum version 3.6.0) to run. The DIA-based PTM localization workflow does not require specially generated spectral libraries and is available in the Spectronaut software tool (minimum version 13).

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

## Acknowledgements

We would like to acknowledge Jan Rudolph for his help in coding and designing the Perseus peptide collapse plugin. Work at The Novo Nordisk Foundation Center for Protein Research (CPR) is funded in part by a generous donation from the Novo Nordisk Foundation (Grant number NNF14CC0001). The proteomics technology developments applied was part of a project that has received funding from the European Union's Horizon 2020 research and innovation programme under grant agreements: MSmed-686547, EPIC-XS-823839, and ERC synergy grant 810057-HighResCells.

## Author contributions

D.B.B.-J. designed and performed all cell line experiments and analyzed the data. O.M.B., L.V., T.G., and L.R. developed the PTM localization algorithm. A.H. developed the Perseus plugin, designed and performed all yeast experiments, and analyzed the data. A.M.-V. performed the experiments for optimizing the phospho-enrichment workflow. C.D.K. performed and analyzed the synthetic phosphopeptide experiments. J.V.O. designed the experiments, critically evaluated the results, analyzed the data, and wrote the manuscript. All authors read, edited, and approved the final version of the manuscript.

## Competing interests

The authors O.M.B., L.V., T.G. and L.R. are employees of Biognosys AG (Zurich, Switzerland). The other authors declare no competing interests.
