## [Peer Review File · Nature Communications]

Reviewers' comments:

Reviewer #1 (Remarks to the Author):

Bekker-Jensen and colleagues have performed DIA proteomics experiments on samples that have been enriched for phosphopeptides. Basic methodological comparisons among DDA and DIA strategies were performed. A site localization algorithm for DIA phosphoproteomics is developed and evaluated, and a stoichiometry assessment strategy is also employed. Perturbational experiments involving EGF stimulation and selected kinase inhibitors are described to demonstrate biological relevance and applicability of the techniques.

This manuscript seems like a “kitchen sink” type of paper, with lots of stuff thrown in that doesn’t really hold together well as a coherent piece of work. The obligatory comparisons between DDA and DIA are favorable to DIA, but this is partially dependent on the care with which they’ve made spectral libraries and the aggressive nature of identifying peptide analytes in the samples.

Major concerns:

The larger number of identifications seem largely drawn from lower abundance peptides, and one might question their quantitative reliability or usefulness. Indeed, after inspecting some of the primary data matrices included as supplemental tables and as direct exports from Spectronaut files, there are many missing values. For example, after exporting quantitative information from the experiment “PilotExp-STYversion13-RPE-library-0-75cutoff-score-1”, related to figure 3, there are 693,918 possible matrix cells (after filters employed by Spectronaut), and over 61% of them (425920) are blank in terms of quantitative measurements. Fewer than 12,000 analytes out of ~36,000 have data present in > 50% of samples. Do the authors need to suggest more stringent filters in the interest of extracting actionable biological information? While the summary statistics in the paper concerning number of analytes identified and quantified seem impressive, is it disingenuous to report these if they can’t yield practical results?

What does adding the directDIA comparisons add? Isn’t making an experiment specific spectral library clearly superior?

Due to the extremely short gradients, most peaks are quantified only with 3-4 points. Would accuracy and completeness be improved with more points under the peak and/or longer gradients? Are some peaks eluting too fast to be reproducibly detected?

What is the basis for picking clusters after use of ANOVA tests to identify “regulated” phosphosites? Are they manually cherry-picked?

Further to that, the authors use the ANOVA test for all their experiments, except for the first Yeast/HELA experiment that represents the “ground truth” It would be very interesting to see the results of the ANOVA test on these samples.

If you are going to do an EGF stimulation experiment, a comparison to the gold-standard paper is warranted. Is it surprising that so few peptides are regulated with such strong perturbations such as EGF stimulation or kinase inhibition? How does this compare to the original SILAC-based EGF stimulation study from Olsen et al., where it was claimed that over 14% of the 6600 phosphosites observed were regulated?

The phosphosite localization scoring algorithm is a strength of this paper. Here are a few recommendations/points of clarification:

In the supplement, the authors indicate that the score is a probability of site localization, except in “toss up” cases where there is no supporting evidence. What is the statistical justification for this? Do the scores follow predictable distributions that can be modeled? It seems more like a fractional intensity of supporting evidence.

It is unclear whether ions that don’t distinguish among localization sites can contribute to the overall score. In attempting to work through an example of the scoring scheme using simulated data, I found that the score would change dramatically based on whether these were included or not.

It might be helpful to include a supplemental spreadsheet that has several worked examples (with numbers from real or simulated data) with simple and complex localization possibilities, so readers can explicitly see how the scores are computed and get a feel for how the score is influenced.

The authors do not describe the cases of phosphopeptides that cannot be distinguished. What is the basis for the arbitrary cutoff of 0.75 localization score? What is the basis for assessing if the phosphopeptide-specific fragment-ion is interfered vs. clean? The elution profile score seems to play a relatively small part that can be overshadowed by abundance.

The authors do not describe the cases when a version of a phosphopeptide is not in the library. Does the algorithm use the expected relative intensities of fragment-ions to identify the peptide? If so, what values does it use for phosphopeptides that are not in the library?

What happens to phosphopeptides that are not localized? Are they counted in the final numbers presented here? Are they used for quantification?

How does the algorithm deal with the following case: A singly phosphorylated peptide with multiple possible sites which is confidently localized in a subset of samples, and present but not-localized in another subset of samples. Does the algorithm consider it as being present and localized in the entire dataset?

To generate figure 4D, were samples dephosphorylated and remeasured in order to generate the stoichiometry measurements? This is not detailed in the methods. Are the data available for this? The exact application of the method should be better described. Overall, the stoichiometry modeling component of the manuscript doesn't add much other than to say "this is possible." The authors also claim that "DIA based stoichiometry estimation is possible with reasonable accuracy" (p.13). What is the value of the accuracy obtained with this method? The authors only provide the mean squared error. More in depth analysis should be done to be able to state this claim.

The authors should compare their results with and cite Thesaurus (Nat Methods. 2019 Aug;16(8):703-706; Searle et al.,2018, BioRxiv), as it presents a very similar tool.

Minor points:

There appear to be two instances of Supplementary Note 1, although they describe different things (scoring, and collapse plugin for perseus).

Supplementary figure 1b needs clarification. What exactly is "centerness" as defined here? How do these graphs support the conclusion that 1 Th overlap is recommended?

Figure 1 c: add labels to venn diagram (e.g. replicate1 vs replicate2)

Figure 1g: label the Y-axis

Figure 1k: Comment on why does dDIA give the highest errors.

Figure 1.l: change legend to properly show the difference of the type of dashed lines

Figure 2: remove "c" misplaced on the left of the figure.

Figure 2 b: define the equation notations. This figure cannot be understood by itself.

Figure 2D: the values on the text do not match the values on the figure.

Supp. figure 2: This figure shows that the score should be bumped to 0.99 to achieve lower error rates when using dDIA. What is the number of phosphopeptides when applying this filter? Does it mean that the filtered signals are of bad quality? It is also unclear if this filter was applied for the results presented in the rest of the publication. Were the numbers of identified phosphopeptides found using this filter? If not this should be made clearer.

Supp. figure 3b. Labels or description is needed. The figure is not understandable by itself without the text.

Supp. figure 3c: Page 10: there is a reference to this figure but it is unclear how the figure illustrates what is said in the text.

Figure 3: c,d: which is DDA and which is DIA? The figure legend and the main text are contradictory.

“Revealed an overall similar pattern...” - Can this be quantitatively measured? How similar are they? Or is it just a visual determination?

Figure 4b: “DDA and DIA estimated the expected ratios closely on median”: Visually this doesn't seem to be the case. Can you provide a quantitative metric that they are indeed close?

Authors state that DIA improve quantification of phosphopeptides at higher intensity. What about the lower intensity? At which point does DIA fail to correctly quantify phosphopeptides?

Figure 4c: The use of the MSE throughout the paper is useful to determine which method can produce the smallest error. While the deconvolution of accuracy and precision is a nice feature of MSE, for the reader, it would be clearer if the actual value of the accuracy and the coefficients of variation would be plotted, as it is a more common metric used in the field.

Figure 4d: the significant/non-significant points are not distinguishable.

Supplementary Figure 4A and 4B: page13: The reference to this figure is not clear. What is said in the text does not match this figure and any other in the study.

Supp. figure 4a: One replicate is missing. Why was it removed? What was the filtering criteria?

Supplementary note 1: Define the notations. Spectronaut (SN), MaxQuant (MQ),...

Unclear if the term “precursor” refers to the case of the same phosphopeptide amino acid sequence with different modified sites, or the same phosphosite measured with different charge states, or different peptides from the same protein.

Figure 5b: unclear what the axis represent.

Figure 5D: Page 15: “found good reproducibility in our dataset”. The authors should provide more quantitative metrics to be able to claim this. How does the figure show this? What is the conclusion of this figure? What information does it provide?

It would be useful to provide a map to the supplemental information in the PRIDE archive, specifically which files belong to which figures (and sub-figures).

Do the MSE calculations and ROC curves include non-phosphopeptides from yeast? If so, this may provide an unrealistic estimation for those whose goal it is to discover regulated phosphopeptides where contamination of non-phosphopeptides may be present.

Was Figure 5A adapted from a source that deserves/requires credit and/or licensing?

Throughout the paper the authors use the terms DDA and DIA to name their experiments. However the experimental conditions change from one experiment to another. In one experiment the “DDA” experiment is done on fractionated samples and not on the other. Maybe the authors should describe in a clearer way the experimental design and use more specific terms as “DDA analysis on fractionated samples”. The legends of the figures should also make this distinction.

How much time does it take to analyze the data with the community-based and the sample-specific libraries?

Reviewer #2 (Remarks to the Author):

The authors present a thorough study to describe the application of data independent acquisition workflows to phosphoproteomic analyses, including the development of new algorithms incorporated into the Spectronaut software to handle localization challenges in DIA spectra. The authors start with method development, followed by algorithm development and testing with synthetic phosphopeptide standards spiked into complex matrices. Furthermore, they compare searching DIA data with various libraries and with a direct search that uses pseudo-MS2 spectra and a standard database search. With these tools in hand, they provide studies examining stoichiometry of phosphorylation modifications and large-scale screenings of thirty kinase inhibitors using their analysis pipeline, showing that large numbers of samples can be screened in a relatively short amount of time to reveal biological relevant results. Overall, this paper is close to being ready to publication, and I recommend for publication following these changes/minor follow up experiments.

Comments:

The biggest change I would like to see is a better demonstration/explanation of the phosphosite localization workflow using an actual peptide example (or several). Currently, Figure 2 and Supplementary Note 1 provide explanation, but no results are shown in the explanation, just theoretical methods for calculation. Given the peptide example in Figure 2a and Supplementary Note 1, can they show an elution group that has confirming ions for the “correct” phosphoisoform and ions that refute incorrect localization sites? Can they match that example with real scores used in the algorithm to give the reader more concrete information about how the algorithm is performing? It would be useful to have one example like this in the main Figure 2 (I would suggest replacing Figure 2c, as it currently does not add much), and having two to three more in the supplementary figures. Something close to this somewhat exists in Supplementary Note 1 (although it is clearly from a library-building run with a 78 min retention time), but it needs to be expanded to specifically show how this calculation functions on real data, and it would be useful to provide an example from the rapid acquisition methods they employ throughout the paper.

In general, I would like to see more information comparing PSMs/phosphopeptides from DDA with elution group precursors and “unique phosphorylated elution group precursors” for DIA. For example, how many fragment ions (and distinctly localizing fragment ions) are matched on average in DIA vs DDA. How confident are sequence assignments in DDA vs DIA now that variable modifications like phosphorylation are added into the equation? Are “unique phosphorylated elution group precursors” truly identified peptide sequences with identified sequencing fragments or just elution groups that match a phosphorylated precursor within an MS1 tolerance without much other confirmatory matching fragments? Perhaps this could go further in explaining why DIA is better at quantifying high intensity phosphopeptides (because they will have a lot of matching fragments)?

Can the authors clarify what they mean by fragment ion count for Figure 1H? Is this signal resulting from all fragment ions? If so, using “signal” would be a far less confusing way to describe this.

Including an explanation of why DIA methods tend to underestimate quantitative ratios (Figure 1j, Supp Figure 1f) compared to DDA would be helpful.

Do the authors have a reason that mean square errors increase in the 2:1 channels for all three methods (Figure 1k)? Also, can the authors spell out why they think DIA underperforms compared to DDA at low intensity ratios?

Is the difference in the stoichiometry calculations between DDA and DIA contributed mainly from phosphopeptide ID and quant, or is it also because of ability to ID and quant the de-phosphorylated peptides?

The data in Figure 1L is very hard to interpret between different conditions, likely because the line coloring and dashing is not appropriate for the size of the figure. I recommend changing the colors or dashing (or both) so there are not so many similar looking lines.

Text in Figure 3c and 3d is far too small to read. I recommend re-scaling other parts of the figure to make the text for the heat maps interpretable. Furthermore, including what the boxes are (merely pointing out similar changes) in the figure legend would be very helpful.

The EasyPhos platform from Humphrey et al. (2015) was developed as a rapid plate-based enrichment for DDA-based phosphoproteomics. I recognize that this paper develops its own shorter DDA method, but citing the EasyPhos paper in context of their method development is important to include.

A minor point is that the authors switch between using quantitation and quantification (and their various derivatives) throughout the paper. I would recommend choosing one and sticking with it throughout the paper for consistency and to minimize confusion.

I believe the figure legend in Figure 1f should read DIA instead of DDA.

Both Supplementary Notes are labeled as Supplementary Note 1, so please change the Perseus plug-in note to Note 2 as it is referred to in the text.

Point-by-point responses to Reviewers' comments

Manuscript number: NCOMMS-19-23955-T

Title: *Rapid and site-specific deep phosphoproteome profiling by data-independent acquisition without the need for spectral libraries*

Our responses to the reviewers' comments are provided below in **blue font**. Key modifications and updates are indicated in the combined manuscript file using track changes.

Reviewer #1 (Remarks to the Author):

Bekker-Jensen and colleagues have performed DIA proteomics experiments on samples that have been enriched for phosphopeptides. Basic methodological comparisons among DDA and DIA strategies were performed. A site localization algorithm for DIA phosphoproteomics is developed and evaluated, and a stoichiometry assessment strategy is also employed. Perturbational experiments involving EGF stimulation and selected kinase inhibitors are described to demonstrate biological relevance and applicability of the techniques.

This manuscript seems like a “kitchen sink” type of paper, with lots of stuff thrown in that doesn't really hold together well as a coherent piece of work. The obligatory comparisons between DDA and DIA are favorable to DIA, but this is partially dependent on the care with which they've made spectral libraries and the aggressive nature of identifying peptide analytes in the samples.

Response: We thank the reviewer for the in-depth revision of our work, and we greatly appreciate that the reviewer acknowledge the care with which we have made the spectral libraries and the power of the 'aggressive' analytical strategy we present. However, we do not agree that this is the reason why DIA is superior to DDA. We demonstrate in the manuscript that direct DIA (dDIA) without spectral libraries is superior to DDA to emphasize that this is NOT dependent on “the care with which we have made spectral libraries”. Moreover, we went a long way to show that site localization, number of IDs, quantitative precision (CVs) as well as quantitative accuracy (fold changes) are advantageous in DIA compared to DDA. We strongly believe that our thorough qualitative as well as quantitative comparisons are unprecedented and that we did everything we could to perform the comparisons in the most fair way and in an unbiased manner.

Major concerns:

The larger number of identifications seem largely drawn from lower abundance peptides, and one might question their quantitative reliability or usefulness. Indeed, after inspecting some of the primary data matrices included as supplemental tables and as direct exports from Spectronaut files, there are many missing values. For example, after exporting quantitative information from the experiment “PilotExp-STYversion13-RPE-library-0-75cutoff-score-1”, related to figure 3, there are 693,918 possible matrix cells (after filters employed by Spectranaut), and over 61% of them (425920) are blank in terms of quantitative measurements. Fewer than 12,000 analytes out of ~36,000 have data present in > 50% of samples. Do the authors need to suggest more stringent filters in the interest of extracting actionable biological information? While the summary statistics in the paper concerning number of analytes identified and quantified seem impressive, is it disingenuous to report these if they can't yield practical results?

Response: We thank the reviewer for pointing this out and we recognize that the DIA methods are not perfect. However, the issues of ‘Missing values’ is inherent to all MS-based quantification methods, no matter whether it is label free DDA, DIA or TMT across multiple blocks. We demonstrate that we have a significant higher overlap between replicates with DIA compared to DDA. To best handle the problem of missing values in the comparison of DIA and DDA, the “actionable biological information” is provided as significant hits after statistical testing. Again, we show that DIA enable identification of more significantly regulated phosphorylation sites and therefore provide more direct biological information about the signaling pathway under investigation.

What does adding the directDIA comparisons add? Isn't making an experiment specific spectral library clearly superior?

Response: We added the direct DIA comparisons to demonstrate that the superiority of DIA compared to DDA is not only due to the high quality of our spectral libraries as also discussed in the answer to the reviewer's first comments above. We agree with the reviewer that making an experiment specific spectral library is clearly superior to direct DIA, but we believe that getting rid of the library generation step would be a big step forward towards a simpler workflow. Library generation is not trivial for every laboratory and it of course generates additional overhead time. Therefore, direct DIA without the need for spectral libraries could make DIA-based phosphoproteomics readily accessible to non-expert laboratories.

Due to the extremely short gradients, most peaks are quantified only with 3-4 points. Would accuracy and completeness be improved with more points under the peak and/or longer gradients? Are some peaks eluting too fast to be reproducibly detected?

Response: We acknowledge that accuracy and completeness may be improved with more points measured under each peak. To that end, a DIA acquisition method can be designed in many different ways with different scan cycle times and isolation window widths. The most important parameter for the method optimization is to achieve the best compromise between peptide identification rates and quantitation accuracy. Importantly, we try to optimize for achieving the best quantitation accuracy no matter with how many points a peak is quantified.

Supplementary figure 1d illustrates that the method chosen results in the lowest CVs. Regarding the question related to longer gradients. We would point to point the reviewer's attention to our previous publication in *Nature Communications* last year (Hogrebe et al. 2018), where we systematically tested the impact of gradient length on phosphopeptide DDA quantification. Here we demonstrated that ultra short gradients actually produced better quantification than longer gradients in terms of statistical significance testing.

What is the basis for picking clusters after use of ANOVA tests to identify "regulated" phosphosites? Are they manually cherry-picked?

Response: We 'picked' the two main clusters based on unsupervised hierarchical clustering of ANOVA significant sites for functional kinase motif analysis based on their regulation profiles. From a biological point-of-view we were mainly interested in the two large clusters of sites upregulated by EGF treatment compared to control. Moreover, one of these two clusters consisted of phosphorylation profiles that were downregulated by the kinase inhibitor treatment, whereas the sites in the other cluster were not affected by the inhibitor treatment..This striking difference allowed us to differentiate between the MEK/ERK-dependent and independent phosphorylation sites, which is exactly the purpose of such an experiment.

Further to that, the authors use the ANOVA test for all their experiments, except for the first Yeast/HELA experiment that represents the "ground truth" It would be very interesting to see the results of the ANOVA test on these samples.

Response: We acknowledge that for consistency it would have been more natural to employ ANOVA tests for all experiments. We opted for the use of the SAM test in the first experiment with the Yeast/Human species mix to generate d-scores, which serve as estimators of t-test statistical significance AND fold change. This can subsequently be used to calculate the ROC curves, similar to what we previously did for DDA data sets comparing different quantitation methods (Hogrebe et al., NCOMMS 2018). However, the ANOVA testing performed in all other experiments were performed with settings, which were essentially the same as for the SAM testing and consequently we are using the same statistical measure already.

If you are going to do an EGF stimulation experiment, a comparison to the gold-standard paper is warranted. Is it surprising that so few peptides are regulated with such strong perturbations such as EGF stimulation or kinase inhibition? How does this compare to the original SILAC-based EGF stimulation study from Olsen et al., where it was claimed that over 14% of the 6600 phosphosites observed were regulated?

Response: We thank the reviewer for highlighting our previous SILAC-based EGF phosphoproteomics paper (Olsen et al., Cell 2006) and we agree that we should compare this to dataset presented here. It is correct that overall 14% of the phosphoproteome in our original SILAC study was regulated by EGF. However, this was based on the analysis of five different EGF stimulation timepoints, where a phosphorylation site was deemed regulated if it was

changing two-fold in at least one of the treatment timepoints. Many of the dynamic sites were only regulated at early (1-5 min) or late (20 min) EGF stimulation times. In the present DIA-based study, we only analyzed a single EGF stimulation time (10 min) and the fraction of regulated sites accounts for roughly 7% of all quantified sites, which is on par with what we observed at 10 min in the SILAC-data.

The phosphosite localization scoring algorithm is a strength of this paper. Here are a few recommendations/points of clarification:

In the supplement, the authors indicate that the score is a probability of site localization, except in “toss up” cases where there is no supporting evidence. What is the statistical justification for this? Do the scores follow predictable distributions that can be modeled? It seems more like a fractional intensity of supporting evidence.

Response: We thank the reviewer for pointing out the importance and strength of the site localization score. We agree with the reviewer that our site localization score can be viewed as a fractional intensity of supporting evidence. It is difficult to calculate the true probabilities for the localization, particularly, when several lines of evidence should be used (in our case feature quality, correlation, absence/presence, etc.). The final localization confidence score (per site) is calculated in a similar fashion to the way it is done in MaxQuant, where the total scores for all Site-Candidates for a given site is summed as calculated as a fraction of the sum of all Site-Candidates. We would therefore use the same established terminology as in MaxQuant and simply call it the site confidence score.

It is unclear whether ions that don't distinguish among localization sites can contribute to the overall score. In attempting to work through an example of the scoring scheme using simulated data, I found that the score would change dramatically based on whether these were included or not.

Response: We are sorry that this was not clear from the text. Since every matched candidate ion can be classified in either confirming or refuting a given site, every possible ion also contributes to the score. Ions that confirm all possible sites therefore also contribute to all Site-Candidates. Moreover, since this is a targeted approach, there can only be ions that are either confirming or refuting. However, there can certainly be peaks in the spectrum that neither nor. But because this is a targeted approach they will not be considered.

It might be helpful to include a supplemental spreadsheet that has several worked examples (with numbers from real or simulated data) with simple and complex localization possibilities, so readers can explicitly see how the scores are computed and get a feel for how the score is influenced.

Response: We thank the reviewer for this suggestion. However, although the algorithm is described in a conceptual manner, the actual algorithm is ~1'400 lines of code. To better explain the different parts in the algorithm we now provide a case example in an excel sheet going through the four steps for one doubly-phosphorylated peptide. This example sheet is now provided as a new Supplementary Table S2, and all other Supplementary table names have been updated accordingly. The new table is described in the manuscript text in the results section about the algorithm, where we added the following sentence:

'An example of the four steps of the site localization algorithm for one doubly-phosphorylated peptide is demonstrated (Supplementary Table S2).'

The authors do not describe the cases of phosphopeptides that cannot be distinguished. What is the basis for the arbitrary cutoff of 0.75 localization score? What is the basis for assessing if the phosphopeptide-specific fragment-ion is interfered vs. clean? The elution profile score seems to play a relatively small part that can be overshadowed by abundance.

Response: The cutoff of 0.75 was chosen because it gave a very similar ratio between true localizations vs false localizations compared to MaxQuant using the same cutoff. A site confidence score 0.75 or higher is generally accepted in the phosphoproteomics field as the cutoff for class1 sites of high-confidence localization (Olsen et al., Cell 2006). Moreover, since the profile correlation score is between -1 and 1 and multiplied with the Log10 of the intensity, any signal must at least correlate positively in order to add to the score. Additionally, mass-shift and isotopic pattern correlation are factored in in a similar fashion. Even the highest abundant signal will not contribute at all if it does not positively correlate. We therefore disagree with the statement that elution profile score seems to play a relatively small part that can be overshadowed by abundance.

The authors do not describe the cases when a version of a phosphopeptide is not in the library. Does the algorithm use the expected relative intensities of fragment-ions to identify the peptide? If so, what values does it use for phosphopeptides that are not in the library?

Response: The PTM localization for library based DIA is still a targeted workflow. This means that for a given peptide version (modified sequence) it will evaluate all possible site options and annotate all of them with a localization confidence score. If all the sites that stem from the peptide assay have a confidence score above the set threshold, the peptide will be considered identified. In other words, peptides/peptidofoms that are not in the library will not be identified. Hence, one also doesn't have the issue of missing relative intensities. Importantly, phosphopeptides not present in the library can still be identified using the direct DIA approach.

What happens to phosphopeptides that are not localized? Are they counted in the final numbers presented here? Are they used for quantification?

Response: Phosphopeptides that are not localized are neither counted in the number of identifications nor are they used for quantification.

How does the algorithm deal with the following case: A singly phosphorylated peptide with multiple possible sites which is confidently localized in a subset of samples, and present but not-localized in another subset of samples. Does the algorithm consider it as being present and localized in the entire dataset?

Response: The PTM localization filter is handled for each peptide/run pair individually. Positive localization is not “carried-over” to the other runs in the experiment. It would therefore cause missing values (NaNs) in those runs where the peptide assay could not be confidently localized (similar to standard Q-value filtering).

To generate figure 4D, were samples dephosphorylated and remeasured in order to generate the stoichiometry measurements? This is not detailed in the methods. Are the data available for this? The exact application of the method should be better described.

Response: We are sorry that this was not clear from the text. For Figure 4D, phospho and non-phospho data from Figure 3 was used, as indicated in the main text. To better clarify this, we have added a sentence in the results section pointing out the data origin. This now reads:

‘This is exemplified by the comparison of EGF versus control samples using the phosphoproteomics dataset from figure 3 based on the project specific library (Figure 4D).’

Overall, the stoichiometry modeling component of the manuscript doesn’t add much other than to say “this is possible.” The authors also claim that “DIA based stoichiometry estimation is possible with reasonable accuracy” (p.13). What is the value of the accuracy obtained with this method? The authors only provide the mean squared error. More in depth analysis should be done to be able to state this claim.

Response: The calculated mean squared errors (MSE) is the sum of positive bias and variance for each method. These represent the quantification error in accuracy and precision, respectively, and thus allow for a direct comparison of these two parameters. The statement should thus be seen relatively compared to the DDA stoichiometry results.

The authors should compare their results with and cite Thesaurus (Nat Methods. 2019 Aug;16(8):703-706; Searle et al.,2018, BioRxiv), as it presents a very similar tool.

Response: We acknowledge this request by the reviewer. To the best of our knowledge, the two algorithms also work in different ways. The purpose of Thesaurus is to find new PTM positional isomers in existing data, whereas our site localization algorithm implemented in Spectronaut is a purely targeted approach. Our algorithm does not look for other isomers in that sense. It only tries to confirm whether a targeted assay is present in the data (by enumerating

all possible site options etc). The workflow presented here will not add new PTM isomers to the results. It only seeks to confirm what was targeted. No matter what, we now cite the Thesaurus algorithm in the context of the challenge of positional isomers. We added the following sentence to the first paragraph of the results section that now reads:

'An additional challenge in DIA is to localize phosphorylation sites correctly and handle positional phosphopeptide isomers (Searle et al., 2019).'

Minor points:

There appear to be two instances of Supplementary Note 1, although they describe different things (scoring, and collapse plugin for perseus).

Response: We agree that this could be confusing and we have therefore renamed them to Supplementary note 1 and 2, respectively. The references are updated in the manuscript.

Supplementary figure 1b needs clarification. What exactly is “centerness” as defined here? How do these graphs support the conclusion that 1 Th overlap is recommended?

Response: To identify the optimal overlap between windows we performed an experiment in which the DIA windows in two consecutive scan cycles were shifted by ‘half-a-window’. This quantitative accuracy is also reflected by the transmission vs centerness, where the transmission is defined as the relation between the intensity of the extreme m/z compared to the intensity of the centermost precursor. The centerness is defined as the relationship between the extreme m/z distance to center and $\frac{1}{2}$ the window size. As supplementary figure 1b shows the method with 1 Th overlap gives the best transmission in the edges of the mass windows. The same conclusion can be reached finding the best compromise between quantitative accuracy and number of identifications (see figures below). In all cases, 1 Th overlap between adjacent DIA windows is the best setting.

Figure 1 c: add labels to venn diagram (e.g. replicate1 vs replicate2)

Response: We added the labels to the venn diagram.

Figure 1g: label the Y-axis

Response: We added label to the y-axis.

Figure 1k: Comment on why does dDIA give the highest errors.

Response: The precision error with dDIA is slightly better than DDA at high intensity ratios, where it is slightly worse at low intensity ratios, which suggests a specific challenge for dDIA. For all ratios except the very low intensity dDIA has the lowest error on accuracy when comparing all three methods, but it is correct that dDIA is challenged at low intensity ratios (0.25:1) compared to standard DIA and DDA. This is likely due to the way dDIA works as it is based on DDA-like database search of the pseudo-MS/MS spectra derived from the DIA analysis. The DIA MS/MS spectra are generally much more complex than DDA-MS/MS, and therefore the identifications by dDIA pseudo-MS/MS rely more on fragment ions of higher abundance.

Figure 1.l: change legend to properly show the difference of the type of dashed lines

Response: We agree with the reviewer that the figure is too complicated and therefore difficult to interpret. Consequently, we have modified the figure such that it only displays two of the dilution ratios instead of three. We also changed the legend accordingly.

Figure 2: remove “c” misplaced on the left of the figure.

Response: We removed the misplaced “c”.

Figure 2 b: define the equation notations. This figure cannot be understood by itself.

Response: We agree that a better explanation of the equations is needed. We have now expanded the figure legend to encompass this. We have added the following explanation:

‘(b) Calculation of site localization confidence score. The score for a given site localization candidate is calculated by summing all confirming fragments matched and subtracting the sum of all refuting fragments matched.’

Figure 2D: the values on the text do not match the values on the figure.

Response: We are sorry for the confusion. The text was indeed not matching the correct figure. We have now updated the manuscript text accordingly to match the figure. The manuscript text now reads:

*‘From the DDA files, we on average correctly localized 108 phosphorylation sites from the synthetic phosphopeptides with 3.1% error rate on wrongly assigned sites (**Figure 2D**). We required at least 0.75 localization probability (Class I sites) as in previous analyses (Olsen et al., 2006). Applying the same score cutoff of 0.75 for the DIA dataset results in correct identification and localization of 153.8 phosphorylation sites on average with 2.8% error rate of incorrectly assigned sites. This indicates that site localization FDR at this cutoff value is comparable to that of DDA analyzed with MaxQuant, but achieving higher site coverage in DIA. Notably, in DDA the number of synthetic phosphopeptide sites identified is significantly hampered at low dilutions. Conversely, DIA maintained a relatively high identification rate across all dilutions, indicating that DIA outperformed DDA in sensitivity and dynamic range (**Figure 2E**).’*

Supp. figure 2: This figure shows that the score should be bumped to 0.99 to achieve lower error rates when using dDIA. What is the number of phosphopeptides when applying this filter? Does it mean that the filtered signals are of bad quality? It is also unclear if this filter was applied for the results presented in the rest of the publication. Were the numbers of identified phosphopeptides found using this filter? If not this should be made clearer.

Response: We thank the reviewer for pointing this out. Based on the synthetic phosphopeptide experiments, the filter of 0.99 is applied for all dDIA experiments throughout the manuscript. This was not clear from the M&M section but is now indicated. The more stringent filter of 0.99 does indeed leads to lower number of identified phosphosites. At the 0.75 cutoff we on average identify 160 sites with dDIA, which is very similar to standard DIA, whereas at the stringent cutoff of 0.99 we identify 143 sites (see figure below). The filter is defined from the same criteria as the filter of 0.75 is defined for spectral library based DIA, which is to obtain a similar ratio of true localized sites and false localized sites compared to MaxQuant. We have now included the figure below as a new Supplementary Figure 2B.

Supp. figure 3b. Labels or description is needed. The figure is not understandable by itself without the text.

Response: We have updated the manuscript text and figure legend to match the figure. The figure legend now reads:

'(b) Overlap of phosphopeptides between the project specific library and the community based library.'

Supp. figure 3c: Page 10: there is a reference to this figure but it is unclear how the figure illustrates what is said in the text.

Response: Thanks for pointing this out. The reference to the figure was wrong and has now been corrected.

Figure 3: c,d: which is DDA and which is DIA? The figure legend and the main text are contradictory.

Response: We edited the text to match the figure.

“Revealed an overall similar pattern...” - Can this be quantitatively measured? How similar are they? Or is it just a visual determination?

Response: The two main clusters based on unsupervised hierarchical clustering of ANOVA significant sites were the same based on their regulation profiles, and the overall clustering

appear visually similar. We therefore used the term 'similar pattern' as it is difficult to directly quantify the similarity of hierarchical clusters.

Figure 4b: "DDA and DIA estimated the expected ratios closely on median": Visually this doesn't seem to be the case. Can you provide a quantitative metric that they are indeed close?

Response: To demonstrate that DIA and DDA correctly estimate the expected ratios we calculated mean +/- SD and median +/- MAD (see table below). These values were calculated for all stoichiometries across all replicates per condition. From the table it is clear that all three methods are able to estimate ratios with low deviations from the expected. Note, DIA performs better than DDA in all cases.

method	target_ratio	mean	standard_deviation	median	median_absolute_deviation
DDA	0.01	0.21	0.24	0.12	0.16
DIA	0.01	0.13	0.17	0.07	0.1
dDIA	0.01	0.11	0.14	0.05	0.08
DDA	0.1	0.15	0.19	0.07	0.07
DIA	0.1	0.12	0.14	0.07	0.06
dDIA	0.1	0.1	0.12	0.06	0.05
DDA	0.5	0.35	0.21	0.32	0.2
DIA	0.5	0.32	0.17	0.32	0.17
dDIA	0.5	0.3	0.17	0.3	0.15
DDA	0.9	0.68	0.25	0.78	0.15
DIA	0.9	0.72	0.23	0.8	0.15
dDIA	0.9	0.73	0.21	0.8	0.12
DDA	0.99	0.72	0.3	0.89	0.14
DIA	0.99	0.76	0.3	0.93	0.1
dDIA	0.99	0.85	0.24	0.97	0.03

Authors state that DIA improve quantification of phosphopeptides at higher intensity. What about the lower intensity? At which point does DIA fail to correctly quantify phosphopeptides?

Response: As shown in figures 1J and 1K, DIA is superior to DDA at all intensities in terms of quantification, whereas dDIA is challenged at lower intensities. This is likely due to the way dDIA works as it is based on DDA-like database search of the pseudo-MS/MS spectra derived from the DIA analysis. The DIA MS/MS spectra are generally much more complex than DDA-MS/MS, and therefore the identifications by dDIA pseudo-MS/MS rely more on fragment ions of higher abundance.

Figure 4c: The use of the MSE throughout the paper is useful to determine which method can produce the smallest error. While the deconvolution of accuracy and precision is a nice feature of MSE, for the reader, it would be clearer if the actual value of the accuracy and the coefficients of variation would be plotted, as it is a more common metric used in the field.

Response: We feel it is subjective what type of analysis method to employ for determining quantitative error rates. In our opinion MSE gives a more in-depth analysis of the quantification over CVs as the MSE also provides an overview of precision rather than only focusing on accuracy. However, we now also calculated the CVs for the different dilutions and as it is clear from the figure below DIA outperforms DDA in all dilutions.

Figure 4d: the significant/non-significant points are not distinguishable.

Response: We have updated the figure such that it should now be easier to distinguish significant from non-significant points.

Supplementary Figure 4A and 4B: page13: The reference to this figure is not clear. What is said in the text does not match this figure and any other in the study.

Response: We thank the reviewer for pointing this out. It seems the Figures were incorrectly referenced in the text and we apologise for this. We have now updated the manuscript text such that it references the correct Supplementary Figures 4c-d.

Supp. figure 4a: One replicate is missing. Why was it removed? What was the filtering criteria?

Response: Unfortunately, we had to remove one replicate sample from the analysis as the raw LC-MS file was 'corrupted' during the recording on the mass spec.

Supplementary note 1: Define the notations. Spectronaut (SN), MaxQuant (MQ),...

Unclear if the term "precursor" refers to the case of the same phosphopeptide amino acid sequence with different modified sites, or the same phosphosite measured with different charge states, or different peptides from the same protein.

Response: We have added the definitions in the introduction to Supplementary note 2. The precursor definition refers to the elution group precursors from Spectronaut (SN) output or PSM precursors from MaxQuant (MQ) evidence output, respectively.

Figure 5b: unclear what the axis represent.

Response: We are sorry for the missing information. The axis represents z-scored and log₂-transformed phosphosite intensities. We have now updated the axis label, so it now says "Phosphosite Log₂ (Z-score)" in the color key.

Figure 5D: Page 15: "found good reproducibility in our dataset". The authors should provide more quantitative metrics to be able to claim this. How does the figure show this? What is the conclusion of this figure? What information does it provide?

It would be useful to provide a map to the supplemental information in the PRIDE archive, specifically which files belong to which figures (and sub-figures).

Response: We agree that it may be difficult to get an overview of the many raw files and associated meta-data. To ease the navigation in the PRIDE archive we now provide a new Supplementary Table S7 in which we describe which files belong to which figures and sub-figures as well as experiments. This information has now been added to the M&M section describing the data availability, which now reads:

*'Data are available via ProteomeXchange with identifier PXD014525. A roadmap linking files in the ProteomeXchange folder to the figures in the manuscript is available (**Supplementary Table S7**).'*

Do the MSE calculations and ROC curves include non-phosphopeptides from yeast? If so, this may provide an unrealistic estimation for those whose goal it is to discover regulated phosphopeptides where contamination of non-phosphopeptides may be present.

Response: No, all MSE calculations and ROC curve analyses are only based on phosphopeptides.

Was Figure 5A adapted from a source that deserves/requires credit and/or licensing?

Response: No, we made the drawing of the signaling network ourselves.

Throughout the paper the authors use the terms DDA and DIA to name their experiments. However the experimental conditions change from one experiment to another. In one experiment the “DDA” experiment is done on fractionated samples and not on the other. Maybe the authors should describe in a clearer way the experimental design and use more specific terms as “DDA analysis on fractionated samples”. The legends of the figures should also make this distinction.

Response: We only fractionate samples for the spectral library generation. All other experiments are performed as single-shot analysis with either DDA or DIA.

How much time does it take to analyze the data with the community-based and the sample-specific libraries?

Response: This depends on the computer, but these were all analyzed with the same computer. This total analysis time was the following: Project specific library: 3.44 h, Combined library: 6.76 h, and directDIA: 40.7 h.

Reviewer #2 (Remarks to the Author):

The authors present a thorough study to describe the application of data independent acquisition workflows to phosphoproteomic analyses, including the development of new algorithms incorporated into the Spectronaut software to handle localization challenges in DIA

spectra. The authors start with method development, followed by algorithm development and testing with synthetic phosphopeptide standards spiked into complex matrices. Furthermore, they compare searching DIA data with various libraries and with a direct search that uses pseudo-MS2 spectra and a standard database search. With these tools in hand, they provide studies examining stoichiometry of phosphorylation modifications and large-scale screenings of thirty kinase inhibitors using their analysis pipeline, showing that large numbers of samples can be screened in a relatively short amount of time to reveal biological relevant results. Overall, this paper is close to being ready to publication, and I recommend for publication following these changes/minor follow up experiments.

Response: We thank the reviewer for his or her positive evaluation of our manuscript and for suggesting publication pending minor changes/additions.

Comments:

The biggest change I would like to see is a better demonstration/explanation of the phosphosite localization workflow using an actual peptide example (or several). Currently, Figure 2 and Supplementary Note 1 provide explanation, but no results are shown in the explanation, just theoretical methods for calculation. Given the peptide example in Figure 2a and Supplementary Note 1, can they show an elution group that has confirming ions for the “correct” phosphoisoform and ions that refute incorrect localization sites? Can they match that example with real scores used in the algorithm to give the reader more concrete information about how the algorithm is performing? It would be useful to have one example like this in the main Figure 2 (I would suggest replacing Figure 2c, as it currently does not add much), and having two to three more in the supplementary figures. Something close to this somewhat exists in Supplementary Note 1 (although it is clearly from a library-building run with a 78 min retention time), but it needs to be expanded to specifically show how this calculation functions on real data, and it would be useful to provide an example from the rapid acquisition methods they employ throughout the paper.

Response: We agree with the reviewer that a better demonstration of the site localization algorithm would be beneficial for the readers. The algorithm is only described in a conceptual manner as the actual algorithm is ~1'400 lines of code. To better explain the different parts in the algorithm we now provide a case example in an excel sheet going through the four steps for one doubly-phosphorylated peptide. This example sheet is now provided as a new Supplementary Table S2, and all other Supplementary table names have been updated accordingly. The new table is described in the manuscript text in the results section about the algorithm, where we added the following sentence:

‘An example of the four steps of the site localization algorithm for one doubly-phosphorylated peptide is demonstrated (Supplementary Table S2).’

In general, I would like to see more information comparing PSMs/phosphopeptides from DDA with elution group precursors and “unique phosphorylated elution group precursors” for DIA. For example, how many fragment ions (and distinctly localizing fragment ions) are matched on average in DIA vs DDA. How confident are sequence assignments in DDA vs DIA now that variable modifications like phosphorylation are added into the equation?

Response: This question can only really be answered for the directDIA case. For library based DIA, the number of fragments for general identification is dependent on the library. We have therefore made a comparison of the identified fragment ions between directDIA (done in Spectronaut) and DDA (done in Spectronaut Pulsar, which is very similar to MaxQuant). We had to do the DDA analysis in Pulsar because we were not able to get sufficient information out of the MaxQuant results. Since the concepts between directDIA and DDA are slightly different, we defined an "Identified Fragment" as any confirming fragment (during the PTM localization process) with a positive score total score. We have plotted the results as boxplot based on the data from the synthetic phosphopeptide dilutions spiked in to the yeast phosphoproteome. To do a fair comparison, the plots only show the overlapping phosphopeptide identifications per sample of the known synthetic phosphopeptides. It is clear from the figure below that the number of fragments used for site localization is very similar between DDA and dDIA, and that this number is generally in the range of 20-40 fragments per MS/MS spectrum.

Note, while a comparison like this can be made, the results can be a bit misleading. That is because this figure only incorporates the aspect that is classically used in PTM localization for DDA. The implementation presented in this paper, however, introduces additional aspects

beyond a mere presence of fragments (like mass shift score, XIC correlation and isotope pattern scoring). Especially in combination with the newly introduced concept of negative evidence (rejecting ions). We therefore decided not to include the figure in the manuscript.

Are “unique phosphorylated elution group precursors” truly identified peptide sequences with identified sequencing fragments or just elution groups that match a phosphorylated precursor within an MS1 tolerance without much other confirmatory matching fragments? Perhaps this could go further in explaining why DIA is better at quantifying high intensity phosphopeptides (because they will have a lot of matching fragments)?

Response: For both peptide-centric (library based) as well as for directDIA, the primary driver for identification are the fragments. This is especially the case for the PTM localization. We guess the question is related to the so-called “Match Between Runs” option in MaxQuant and whether the DIA results presented here were processed in such a manner. The answer is no, all identifications (and PTM localizations) stand on their own.

Can the authors clarify what they mean by fragment ion count for Figure 1H? Is this signal resulting from all fragment ions? If so, using “signal” would be a far less confusing way to describe this.

Response: The fragment ion count is the “RawOvFtT”, which is a parameter that can be read out from the individual scan headers in the raw files. It represents the total number of ions analyzed in the orbitrap for the given scan. To clarify this better, we have added the following label to the y-axis in Figure 1H: *‘Ions measured in the orbitrap (MS2)’*

Including an explanation of why DIA methods tend to underestimate quantitative ratios (Figure 1j, Supp Figure 1f) compared to DDA would be helpful.

Response: This is an interesting observation made by the reviewer. It looks like DDA is slightly overestimating the ratios, whereas DIA is slightly underestimating the ratios. A likely explanation for this could be the difference in how the quantification is performed. DDA is based on full-scan MS1 quantification where we always reach the preset AGC target value, whereas quantification in DIA is performed on MS/MS-level, where we typically do not reach the preset AGC target value within the maximum allowed injection time for each DIA scan.

Do the authors have a reason that mean square errors increase in the 2:1 channels for all three methods (Figure 1k)? Also, can the authors spell out why they think DIA underperforms compared to DDA at low intensity ratios?

Response: We believe that the reason for the increased mean squared errors in the 2:1 channels compared to the 1.5:1 channel is that mass spectrometric quantification is best at determining ratios that are close to 1:1. It is correct that dDIA is challenged at low intensity

ratios (0.25:1) compared to standard DIA and DDA. This is likely due to the way dDIA works as it is based on DDA-like database search of the pseudo-MS/MS spectra derived from the DIA analysis. The DIA MS/MS spectra are generally much more complex than DDA-MS/MS, and therefore the identifications by dDIA pseudo-MS/MS rely more on fragment ions of higher abundance.

Is the difference in the stoichiometry calculations between DDA and DIA contributed mainly from phosphopeptide ID and quant, or is it also because of ability to ID and quant the de-phosphorylated peptides?

Response: Yes, we do believe it is a combined effect since DIA will improve quantification of both. We have previously demonstrated that DIA analysis with short gradients increase peptide coverage by a factor of two or more while maintaining quantitative accuracy (Kelstrup et al., J. Proteome Res. 2018).

The data in Figure 1L is very hard to interpret between different conditions, likely because the line coloring and dashing is not appropriate for the size of the figure. I recommend changing the colors or dashing (or both) so there are not so many similar looking lines.

Response: We agree with the reviewer that the figure is too complicated and therefore difficult to interpret. Consequently, we have modified the figure such that it only displays two of the dilution ratios instead of three. We also changed the legend accordingly.

Text in Figure 3c and 3d is far too small to read. I recommend re-scaling other parts of the figure to make the text for the heat maps interpretable. Furthermore, including what the boxes are (merely pointing out similar changes) in the figure legend would be very helpful.

We agree that the text was too small and have now increased the font size accordingly. We have also added one sentence of additional explanation in the figure legend for Figure 3e, that now reads:

'(e) Linear sequence motif analysis for two major clusters, marked in colored boxes on heatmaps.'

The EasyPhos platform from Humphrey et al. (2015) was developed as a rapid plate-based enrichment for DDA-based phosphoproteomics. I recognize that this paper develops its own shorter DDA method, but citing the EasyPhos paper in context of their method development is important to include.

Response: Thanks for this suggestion, which we have incorporated in the discussion of our method in the first paragraph of the discussion section that now reads:

'While it has been shown previously that rapid plate-based enrichment for DDA-based sensitive phosphoproteomics starting from 200 ug or less (Humphrey et al., 2018; Post et al., 2017) is feasible, this study represents an important advancement in that we use sensitive phosphoproteomics to analyze cellular signaling networks much faster and with greater depth.'

A minor point is that the authors switch between using quantitation and quantification (and their various derivatives) throughout the paper. I would recommend choosing one and sticking with it throughout the paper for consistency and to minimize confusion.

Response: We agree that we should be consistent. We only use the term 'quantification' and 'quantify' throughout the text. The only exception is the term 'quantitative' for which no other good alternative is available.

I believe the figure legend in Figure 1f should read DIA instead of DDA.

Yes, it is now corrected

Both Supplementary Notes are labeled as Supplementary Note 1, so please change the Perseus plug-in note to Note 2 as it is referred to in the text.

Corrected

Reviewers' comments:

Reviewer #1 (Remarks to the Author):

I thank the authors for their detailed responses to my review comments. I wish that more of their responses had found their way into the manuscript itself to help readers judge the method for themselves. Based on their responses, I will now be more explicit about the things that I believe should be included as part of the manuscript.

As well, the authors seem to be putting a lot of stock in the direct DIA (dDIA) method as “DIA for non-expert users”. Apologies for not catching this the first time around, but if this is the first publication to describe the method then we need to see some more details other than: “In this approach, spectral libraries are generated directly by searching deconvoluted pseudo MS/MS spectra from DIA data against a peptide database. For this process, Pulsar, the search engine in Spectronaut, applies the same search settings as DDA searches in MaxQuant.” My literature searching did not turn up any prior publications thoroughly describing the dDIA method, except a technical note from Biognosys. DIAUmpire (a similar technology) is not even referenced. And regarding the comparison to DIA to dDIA, I think you should state in the discussion that “regular” DIA will still be superior to direct DIA.

Regarding missing values, for all methods in all experiments, provide tables showing: total number of possible values measured, total numbers actually measured, and the % of missing values.

Regarding directDIA comparison to DIA, add qualifying information to the discussion stating that: “making an experiment specific spectral library is clearly superior to direct DIA,” as per the rebuttal letter.

Regarding # of points across peaks, provide supplemental figures showing the distributions of points across the peaks for quantitative measurements for all methods in all experiments.

Regarding clusters picked in Fig 3: provide the row dendrogram so that the reader can see that the clusters selected are actually the two largest and were done in a data driven manner.

Provide ANOVA analysis of first yeast/HeLa experiment in the supplement.

Regarding comparison of original SILAC EGF stimulation experiments to the current ones, statement of numbers of regulated peptides is not enough. Select the relevant timepoint(s) from the original Olsen et al. publication and compare the actual biological results and insights gained. This would go a long way to inspiring readers' confidence in the methods presented.

Remove/change all references to the site localization score as a "probability." You agreed in your response that it is not a probability.

For the worked example, provide the actual spectra and/or extracted ion chromatograms used to score the example. State in methods that non-localizing ions are not considered for the localization score. But if non-localizing ions can't contribute to the score, why are the b1 and b2 ions not marked as excluded?

Explicitly state that a phosphorylation configuration must be present in the library for it to be considered (as opposed to Thesaurus), and that non-localized sites are not considered as identifications for purposes of the studies you present.

Qualify the statements about accuracy of stoichiometry prediction as being "relative" to other approaches but not absolute. State that no absolute consideration of stoichiometry accuracy was performed here nor could be derived from these data.

Provide explanation and supplemental figure on "centerness" as you have provided in your rebuttal in the manuscript itself.

Include discussion of why dDIA does worse than DDA at low abundance ratios in the paper.

Include the supplemental table from your rebuttal regarding mean \pm SD and median \pm MAD as a table in the manuscript.

Include the version of Figure 4C with CVs from your rebuttal as a supplemental figure.

Reviewer #2 (Remarks to the Author):

The authors responded to a majority of my concerns. I am glad to see the inclusion of the example of localization, and I understand their case for not including the number of fragment ions matched between DDA and directDIA in the manuscript. Below I include points that came up as I read their responses and that will be valuable to be included in the final version of the manuscript.

The authors make a good point in their response about DIA underestimating ratios because of quantitation differences made between DIA and DDA being from MS2 vs MS1 scans, respectively. First, I would like to see their explanation included in the text (near the discussion of Figure 1J, page 8 lines 163-167). Secondly, this brings up an interesting implication for the comparison they make in Figure 1G and 1H. A major point here is that an AGC target of $3e6$ was used for MS2 scans in DIA experiments compared to $1e5$ for DDA MS2s. Although the injection times were set to be the same, it makes sense that the DIA MS2 scans would have significantly more signal by default, even if DIA scans always maxed their injection times (even presuming that some DDA MS2 scans hit their target without hitting their maximum injection times). This difference in method set up warrants a caveat in the discussion of Figure 1G and 1H to acknowledge the difference between ion targets that could account for this difference in signal.

Furthermore, the authors plot ions for MS2 scans for DDA and DIA, but the MS2 scans are only used for quantification for DIA. The authors explain that quantitative reproducibility is better in DIA, which is likely due to the more efficient use of the ion beam for MS/MS scans (page 8, lines 147-153). But the MS2 level is of course not where DDA experiments derive their quantitative information in this work. The reproducibility must come from the number of measurements over an elution profile (i.e., multiple fragments) in DIA compared to the single elution profile quantitation, instead of the signal in the MS2 scans alone, right? For example, if just quantifying DIA identifications using the most abundant fragment (or two), surely the quantitative reproducibility drops to match or be below DDA. I would like to see the authors address this in the text to give a more accurate representation of what is contributing to these gains in identifications vs. reproducibility.

Point-by-point responses to Reviewers' comments

Manuscript number: NCOMMS-19-23955A

Title: *Rapid and site-specific deep phosphoproteome profiling by data-independent acquisition without the need for spectral libraries*

Our responses to the reviewers' comments are provided below in **blue font**. Key modifications and updates are indicated in the combined manuscript file using track changes.

Reviewer #1 (Remarks to the Author):

I thank the authors for their detailed responses to my review comments. I wish that more of their responses had found their way into the manuscript itself to help readers judge the method for themselves. Based on their responses, I will now be more explicit about the things that I believe should be included as part of the manuscript.

Response: We thank the reviewer for his or her positive comments on our revised manuscript. We agree that it would strengthen our manuscript even more to incorporate some of the details from the rebuttal letter. We have therefore followed the advice of the reviewer and updated the manuscript accordingly. See point-by-point responses below.

As well, the authors seem to be putting a lot of stock in the direct DIA (dDIA) method as “DIA for non-expert users”. Apologies for not catching this the first time around, but if this is the first publication to describe the method then we need to see some more details other than: “In this approach, spectral libraries are generated directly by searching deconvoluted pseudo MS/MS spectra from DIA data against a peptide database. For this process, Pulsar, the search engine in Spectronaut, applies the same search settings as DDA searches in MaxQuant.” My literature searching did not turn up any prior publications thoroughly describing the dDIA method, except a technical note from Biognosys. DIAUmpire (a similar technology) is not even referenced. And regarding the comparison to DIA to dDIA, I think you should state in the discussion that “regular” DIA will still be superior to direct DIA.

Response: We thank the reviewer for pointing this out and we agree that the direct DIA strategy needs more explanation including a reference to the DIA-Umpire manuscript. Accordingly, we have now included an additional paragraph in the methods section explaining the direct DIA approach in more detail. The paragraph reads:

Direct data-independent acquisition (dDIA) analysis strategy in Spectronaut

To search DIA data directly against a protein database FASTA file, the DIA raw dataset is first converted into DDA-like data of individual precursors and their associated fragment peak lists (DIA-Umpire Tsou et al. 2015). To generate such pseudo-DDA data, an MS1 feature detection

is performed in three-dimensions (m/z, intensity, and retention time). The resulting 3D features are created by connecting scan-wise 2D features. Finally, precursor-fragment groups are created by prioritizing fragment ions corresponding to the individual 3D precursor features. The pseudo-DDA data is then searched using the Pulsar search algorithm, which is a classical peptide search engine similar to Andromeda in MaxQuant and Mascot (ref. Andromeda, Mascot). This approach allows searching complex DIA data in a similar fashion as classical DDA data without the need of a spectral library.'

We have also added a sentence to the manuscript text in which we state that regular DIA is superior to direct DIA as requested by the reviewer by adding the following sentence to the last paragraph of the discussion section:

'Although classic DIA analysis with an experiment specific spectral library is superior to dDIA, it is much easier to implement the dDIA workflow making it easier accessible to the general proteomics community'

Regarding missing values, for all methods in all experiments, provide tables showing: total number of possible values measured, total numbers actually measured, and the % of missing values.

Response: We have now included an additional table as a new tab in Supplementary table S3 named 'data completeness' in which we indicate the percent of missing values for all the all the five analysis strategies compared: (1) DDA, (2) DIA with project specific library, (3) DIA with community based library, (4) DIA with both libraries combined and (5) direct DIA.

Regarding directDIA comparison to DIA, add qualifying information to the discussion stating that: "making an experiment specific spectral library is clearly superior to direct DIA," as per the rebuttal letter.

Response: We have followed the reviewer's advice and incorporated his suggestion of stating that DIA is superior to directDIA by adding the following sentence to the last paragraph of the discussion section:

'Although classic DIA analysis with an experiment specific spectral library is superior to dDIA, it is much easier to implement the dDIA workflow making it easier accessible to the general proteomics community'

Regarding # of points across peaks, provide supplemental figures showing the distributions of points across the peaks for quantitative measurements for all methods in all experiments.

Response: We thank the reviewer for this suggestion as we agree that this is an important measure to include for evaluating the quantitative accuracy of our acquisition methods. Accordingly, we extracted the information about how many data points across the elution profiles for all precursors identified with the different DIA acquisition methods. We visualized the results as box-plots for each acquisition method in a new Supplementary figure S1F (see insert below). From this analysis, it is clear that the short cycle-times – which are the methods we use throughout the manuscript - provide the highest number of data points per peak and hence best quantification.

This new supplementary figure S1F is discussed in the first part of the result section to which we added the following sentences:

‘Another important measure for evaluating quantitative accuracy of DIA acquisition methods is the number of data points measured across the elution profiles for precursors identified. Analyzing this for the different DIA acquisition methods reveals that the shortest DIA scan cycle time of 2 seconds provides the highest number of data points per peak and hence best quantification (Supplementary Figure 1F).’

Regarding clusters picked in Fig 3: provide the row dendrogram so that the reader can see that the clusters selected are actually the two largest and were done in a data driven manner.

Response: We thank the reviewer for pointing this out. We have now included the row dendrograms to all five heatmaps of the hierarchical clustering presented in Fig.3 and Supplementary Fig. 3.

Provide ANOVA analysis of first yeast/HeLa experiment in the supplement.

Response: We now performed the ANOVA analysis requested by the reviewer, and made a list/figure of the results, which is provided as a new Supplementary figure S11. Accordingly, we also updated the methods section describing the ANOVA analysis with the following paragraph:

'For the analysis of biological benchmark data and the kinase inhibitor screen, the Spectronaut reported normalized intensity data matrices were transformed into phosphorylation site tables with the plugin peptide collapse script using localization cutoff of 0.75. Downstream data analysis was performed by filtering for minimum of three valid values in at least one treatment group followed by median subtraction across conditions. Missing values were imputed by random sampling of the lower end of the normal distribution (width 0.3 and downshift 1.8) in total matrix. ANOVA significance testing was performed on z-scored log-transformed phosphorylation site intensities with parameter settings of $s_0=0.1$ and $FDR=0.05$ or 0.01 . Heatmaps of all significantly regulated phosphorylation sites were generated by unsupervised hierarchical clustering using Pearson correlation analysis.'

We describe the new supplementary figure S11 at the end of the first part of the results section, where we added the following sentence:

'Likewise, an analysis of variance (ANOVA) statistical test revealed a higher number of significantly regulated yeast phosphopeptides for the DIA methods compared to DDA (Supplementary Figure 11).'

Regarding comparison of original SILAC EGF stimulation experiments to the current ones, statement of numbers of regulated peptides is not enough. Select the relevant timepoint(s) from the original Olsen et al. publication and compare the actual biological results and insights gained. This would go a long way to inspiring readers' confidence in the methods presented.

Response: We thank the reviewer for suggesting to include a more thorough comparison to our original SILAC-based EGF phosphoproteomics study (Olsen et al., Cell 2006). Accordingly, we have now overlapped the two datasets focusing on the relevant stimulation time-points from the

original publication and compared the actual biological results and insights gained. We find that DIA covers more than two-fold more of the EGF-regulated HeLa sites compared to the DDA data, and that these sites belong to the clusters of phosphorylation site kinetics with maximum phosphorylation at 10 minutes of EGF stimulation. Moreover, unbiased pathway-enrichment analysis of the DIA significant sites also found in HeLa cells revealed EGFR1 signaling pathway as the most overrepresented pathway, which independently validates the quality of the DIA-based quantification. We have now added the following paragraph to the end of the results section discussing Figure 3 and included a new Supplementary Table S4 to discuss this comparison:

*Correct biological interpretation of phosphoproteomics data is dependent of the precision and accuracy of quantification for the identified phosphopeptides (Hogrebe et al., NCOMMS 2018). To evaluate the quality of label-free quantification methods for providing insights into EGF-signaling, we benchmarked our DIA and DDA datasets against a gold-standard reference dataset of EGF-dependent phosphorylation sites dynamics (Olsen et al., Cell 2006). Of the 1050 dynamically regulated phosphorylation sites previously identified by SILAC-based quantitation of five EGF stimulation time points of HeLa cells (Olsen et al, Cell 2006, Suppl. Table 6), we covered 597 of them in our label-free DIA analysis and 504 of these sites in the DDA analysis, respectively. Note, that we analyzed RPE1 cells in the current study and not HeLa cells, and perfect overlap is therefore not to be expected due to significant differences in protein abundances and hence phospho-signaling network activities between cell lines (Bekker-Jensen et al, Cell Systems 2017). We only analyzed a single EGF stimulation time (10 min) in RPE1 cells and the fraction of regulated sites accounts for roughly 7% of all quantified sites, whereas many of the dynamic sites in HeLa cells were regulated at early (1-5 min) or late (20 min) EGF stimulation times. Of the overlapping EGF-dependent HeLa sites, 189 were ANOVA significant in the DIA analysis, whereas only 81 of them were ANOVA significant in the DDA analysis emphasizing the power of DIA for covering more relevant sites. As anticipated, the ANOVA significant DIA sites were mainly upregulated in HeLa cells after 10 minutes EGF stimulation with an average log₂ fold-change of 1.0 and belonging to the clusters B, C and D of intermediate stimulators, late stimulators, and terminal effectors defined in the HeLa dataset (Olsen et al., Cell 2006). Conversely, the non-significant sites had an average log₂ fold-change after 10 minutes EGF of 0.5, and they mainly belonged to clusters E and F of early negative regulators and late negative regulators with maximum phosphorylation site changes at 20 minutes of EGF stimulation. Performing a pathway-enrichment analysis of the 189 significant sites from the DIA dataset revealed the EGFR1 signaling pathway as the most significantly overrepresented pathway ($p=1.57E-11$ and Benjami-Hochberg correct $p=6.06E-09$) among the 746 distinct pathways covered (**Supplementary Table S4**). This benchmark underlines the high quality of DIA-based quantification for revealing biological insights.*

Remove/change all references to the site localization score as a “probability.” You agreed in your response that it is not a probability.

Response: We agree that our site localization score is not strictly a probability score. As described in the rebuttal, manuscript text and the supplementary note, the localization candidate scores are condensed into a Site Confidence Score (value between 0 and 1 for each site) in a

similar way as MaxQuant does with the Andromeda score. Accordingly, we have updated the manuscript text and replaced 'probability' with 'site confidence score'.

For the worked example, provide the actual spectra and/or extracted ion chromatograms used to score the example. State in methods that non-localizing ions are not considered for the localization score. But if non-localizing ions can't contribute to the score, why are the b1 and b2 ions not marked as excluded?

Response: Following the reviewer's request, we now provide an annotated MS/MS spectrum and extracted fragment ion chromatogram for the example data. We have incorporated them into the manuscript as new supplementary figures S2A and S2B.

Note that the b1 and b2 are not marked as "excluded" because those ions also contribute to the scoring. The idea being that ions that contribute to all localization candidates will cancel each other out in the final score (since they contributed to all candidates equally). The flag "excluded" is only used for fragments that have conflicting states of evidence (like b3-H₂O because that particular ion is indistinguishable from phospho-Serine with neutral loss of phosphoric acid or regular Serine with H₂O-loss, and would therefore neither constitute positive nor negative evidence). The PTM localization algorithm deals with ions that both confirm and refute a specific site.

Explicitly state that a phosphorylation configuration must be present in the library for it to be considered (as opposed to Thesaurus), and that non-localized sites are not considered as identifications for purposes of the studies you present.

Response: We thank the reviewer for pointing this out. We have now incorporated a paragraph about this in the discussion section that now reads:

'In a classic DIA analysis, a phosphorylation site must be present in the library for it to be considered. Conversely, in dDIA all possible phosphorylation site combinatorics for a given peptide is considered similar to DDA.'

Qualify the statements about accuracy of stoichiometry prediction as being "relative" to other approaches but not absolute. State that no absolute consideration of stoichiometry accuracy was performed here nor could be derived from these data.

Response: We agree with the reviewer that no absolute stoichiometry estimates are performed and accordingly we have deleted the mentioning of analysis of absolute occupancy from the text.

Provide explanation and supplemental figure on "centerness" as you have provided in your rebuttal in the manuscript itself.

Response: To follow the suggestion made by the reviewer, we now provide the requested figure as a new Supplementary Figure 1B and detailed explanation on ‘centerness’ concept in the results section that now reads:

*‘The best compromise between quantitative accuracy and number of identifications is reached with 1 Th overlap (**Supplementary Figure 1B**). As an alternative strategy to identify the best overlap, we analyzed the relationship between ion transmission and centerness, where the transmission is defined as the relation between the intensity of the extreme m/z compared to the intensity of the centermost precursor for DIA windows in two consecutive scan cycles shifted by half-a-window. The centerness is defined as the relationship between the extreme m/z distance to center and ½ the window size.’*

Include discussion of why dDIA does worse than DDA at low abundance ratios in the paper.

Response: We agree that this is valuable to discuss in the manuscript and have added the following sentences to the results section describing the quantification accuracy of DDA and dDIA:

‘This is likely due to the way dDIA works as it is based on DDA-like database search of the pseudo-MS/MS spectra derived from the DIA analysis. The DIA MS/MS spectra are generally much more complex than DDA-MS/MS, and therefore the identifications by dDIA pseudo-MS/MS rely more on fragment ions of higher abundance.’

Include the supplemental table from your rebuttal regarding mean +- SD and median +- MAD as a table in the manuscript.

Include the version of Figure 4C with CVs from your rebuttal as a supplemental figure.

Response: We have added the requested items to the manuscript as new Supplementary Figures 4A and 4B, which we describe in the results section with the following paragraph:

*‘To demonstrate that DIA and DDA correctly estimate the expected ratios, we calculated mean standard deviations and median absolute deviations for each stoichiometry mix (**Supplementary Figure 4A**). These values were calculated for all stoichiometries across all replicates per condition. Together with the calculated CVs for the different dilutions it is clear that DIA outperforms DDA in all dilutions (**Supplementary Figure 4B**).’*

Reviewer #2 (Remarks to the Author):

The authors responded to a majority of my concerns. I am glad to see the inclusion of the example of localization, and I understand their case for not including the number of fragment ions matched between DDA and directDIA in the manuscript. Below I include points that came up as I read their responses and that will be valuable to be included in the final version of the

manuscript.

Response: We thank the reviewer for his or her positive comments on our revised manuscript. We have also followed the advice of the reviewer and updated the manuscript with the remaining points requested. See point-by-point responses below.

The authors make a good point in their response about DIA underestimating ratios because of quantitation differences made between DIA and DDA being from MS2 vs MS1 scans, respectively. First, I would like to see their explanation included in the text (near the discussion of Figure 1J, page 8 lines 163-167).

Response: We agree that this would be valuable to add to the manuscript text. Accordingly, we have included the following paragraph in the results section:

'From the box-plots it looks like DDA is slightly overestimating the ratios, whereas DIA is slightly underestimating the ratios. A likely explanation for this could be the difference in how the quantification is performed. DDA is based on full-scan MS1 quantification, where the preset target value is usually reached. Conversely, quantification in DIA is performed on MS/MS-level, where the preset target value is often not reached within the maximum allowed injection time for each DIA scan.'

Secondly, this brings up an interesting implication for the comparison they make in Figure 1G and 1H. A major point here is that an AGC target of $3e6$ was used for MS2 scans in DIA experiments compared to $1e5$ for DDA MS2s. Although the injection times were set to be the same, it makes sense that the DIA MS2 scans would have significantly more signal by default, even if DIA scans always maxed their injection times (even presuming that some DDA MS2 scans hit their target without hitting their maximum injection times). This difference in method set up warrants a caveat in the discussion of Figure 1G and 1H to acknowledge the difference between ion targets that could account for this difference in signal.

Response: As requested, we now discuss the difference in target values for MS/MS between DIA and DDA. We have added the following sentence to the discussion of Figure 1G and 1H:

'This difference is to be expected as target of $3e6$ was used for MS/MS scans in DIA experiments compared to target value of $1e5$ for DDA MS/MS.'

Furthermore, the authors plot ions for MS2 scans for DDA and DIA, but the MS2 scans are only used for quantification for DIA. The authors explain that quantitative reproducibility is better in DIA, which is likely due to the more efficient use of the ion beam for MS/MS scans (page 8, lines 147-153). But the MS2 level is of course not where DDA experiments derive their quantitative information in this work. The reproducibility must come from the number of measurements over an elution profile (i.e., multiple fragments) in DIA compared to the single elution profile quantitation, instead of the signal in the MS2 scans alone, right? For example, if just quantifying DIA identifications using the most abundant fragment (or two), surely the quantitative

reproducibility drops to match or be below DDA. I would like to see the authors address this in the text to give a more accurate representation of what is contributing to these gains in identifications vs. reproducibility.

Response: We agree with the reviewer that the higher reproducibility in DIA-based quantification must be due to measurements of multiple fragment ions, and hence better ion statistics than DDA. Accordingly, we have included the following paragraph in the results section discussing this point:

'Moreover, the higher reproducibility of DIA compared to DDA likely comes from the multiple fragments measured over an elution profile, whereas single elution profile quantification is performed in DDA.'